# Urban Rivers Corridors in the Don Catchment, UK: From Ignored, Ignoble and Industrial to Green, Seen and Celebrated

**Ed Shaw \*, Debbie Coldwell, Anthony Cox, Matt Duffy, Chris Firth, Beckie Fulton, Sue Goodship, Sally Hyslop, David Rowley, Rachel Walker and Peter Worrall**

Don Catchment Rivers Trust, Churchill Business Centre, Churchill Road, Doncaster DN2 4LP, UK; debbie.coldwell@dcrt.org.uk (D.C.); anthony.cox@dcrt.org.uk (A.C.); matt.duffy@dcrt.org.uk (M.D.); chris.firth@dcrt.org.uk (C.F.); beckie.fulton@dcrt.org.uk (B.F.); sue.goodship@dcrt.org.uk (S.G.); sally.hyslop@dcrt.org.uk (S.H.); david.rowley@dcrt.org.uk (D.R.); rachel.walker@dcrt.org.uk (R.W.); p.worrall@blueyonder.co.uk (P.W.)
\* Correspondence: ed.shaw@dcrt.org.uk; Tel.: +44-1302-439081

**Abstract:** Research on urban rivers often seeks to find commonalities to advance knowledge of the effect of urbanisation on rivers, and rightly so. But it is important, also, to develop a complementary understanding of how urban rivers can be distinct, to facilitate a more nuanced view of concepts such as the 'urban river syndrome' and of the challenges facing those who wish to create more sustainable urban river corridors. To this end we use the Don Catchment as a case study to illustrate how historic patterns of urbanisation have been fundamental in shaping the catchment's rivers. Following the Industrial Revolution, the catchment became an industrial centre, resulting in the ecological death of river ecosystems, and the disconnection of communities from stark urban river corridors. Widescale deindustrialisation in the 1970s and 1980s then resulted in a partial ecological recovery of the rivers, and ignited public interest. This history has imbued the catchment's urban river corridors with a distinctive industrial character that can vary greatly between and within settlements. It has also left a legacy of particular issues, including a high degree of river habitat fragmentation and physical modification, and of negative perceptions of the rivers, which need improving to realise their potential as assets to local communities.

**Keywords:** urban river corridor; urban rivers; urban streams; urban ecology; industrialisation

## 1. Introduction

Long have rivers been important sources of food, water, materials, power and transport to societies around the world [1,2], and as a result, they have served as focal features in the landscape around which settlements and civilisations have developed. As human populations have rapidly expanded and urbanised in the modern era, ever greater lengths of river have become absorbed and confined within the urban fabric, putting urban rivers and streams at the heart of many towns and cities across the world.

The fact that many human populations live in close proximity to urban rivers means that the potential ecosystem services and disservices that they provide are very important [3]. It is therefore unsurprising that in recent decades they have attracted considerable investment to enhance the benefits they provide to society [4,5].

There is a rapidly growing body of research on urban rivers [5], and a key concept that has emerged from this work is that of 'urban river syndrome'; arising from the finding that urbanisation commonly degrades river ecosystems across a common suite of physical and ecological variables [6]. The symptoms of the syndrome include a flashier hydrograph, declines in water quality, modified channel morphology, and reduced biodiversity [6].

Yet, not all instances of urban river syndrome are the same, as variations in the character of urban areas, such as the extent of impervious surfaces, the nature of the dominant human activities, and historic patterns of development, interact with natural

factors such as climate, and place a particular suite of pressures on river ecosystems [5,7]. This means that there is no one best treatment to undo the effect of the syndrome [5]. Furthermore, different communities and stakeholders have differing needs, values and interests, and so the most appropriate target restoration state for a river needs tailoring to the local context, and may be different from its pre-urban state [8]. Effective decision making is therefore dependent on a nuanced appreciation of the unique combination of historic and current ecological, cultural, social and economic factors that have shaped these rivers.

The authors are affiliated with the Don Catchment Rivers Trust, an environmental charity that delivers projects to improve the ecological condition of the rivers and streams of the Don Catchment, a particularly urbanised catchment in northern England, UK. Over a period of centuries, urbanisation has completely transformed most of the rivers and streams in the catchment. This transformation can be divided into two phases; (1) industrialisation and urbanisation of the catchment, the degradation of the river ecosystems, the decline of ecosystem service provision, and the disconnection of urban dwellers with the rivers; and (2) deindustrialisation, partial river ecosystem recovery, and increasing recognition and celebration of the rivers as assets for local communities.

A guiding principle of the Trust's works is to place the local community at the core of river conservation. It is our view that by offering new opportunities for people to interact with their rivers, lasting connections can be built between people and place, and the community empowered to protect urban rivers—a crucial step in realising their potential as green-blue infrastructure.

In this paper we outline these phases, reflecting on the cultural, social, and environmental processes that have shaped the urban rivers of the Don Catchment and people's relationships with them. We describe three main elements to the Trust's work; restoring riverine ecological connectivity, Natural Flood Management, and reconnecting people to the rivers, and reflect on how the industrial past has influenced our activities. Lastly, we examine the previous topics to discuss the specificity of the Don Catchment's urban river corridors.

## 2. Catchment Overview

The River Don rises in the moorland of the uplands of the Pennines before heading more or less eastwards through a string of settlements, including the city of Sheffield, and the large towns of Rotherham and Doncaster (see Figure 1). After passing through these urban areas, the Don cuts across intensive arable lowlands, and discharges into the River Ouse a little upstream of the Humber Estuary. The Don has two main tributaries; the Rivers Rother and Dearne. The Rother drains the south of the catchment, and flows through the town of Chesterfield, while the Dearne drains the north of the catchment, and passes through the town of Barnsley.

Together the Don, Dearne, Rother, and their many tributaries drain a moderately sized area of northern England (~1700 km$^2$). However, by global standards the rivers are small. The mean flow of the Don at the UK Environment Agency Doncaster gauging station is 16 m$^3$/s, where the river is about 20 m wide. Compare this, for instance, with the Thames, with a mean discharge of 65 m$^3$/s [9], and a width of ~250 m in central London; the Rhine, with a mean discharge of 2330 m$^3$/s [9], and a width of ~400 m in Rotterdam, or the Yangtze, with a mean discharge of 31,900 m$^3$/s [9], and a width of 1.5 km or more at Nanjing.

The Don Catchment contains the Sheffield Urban Area conurbation, which is the 8th largest in the UK, and for its size the catchment is relatively urban, comprising 18% of the landuse by area [10]. The population of the catchment stands at approximately 1.39 million [11].

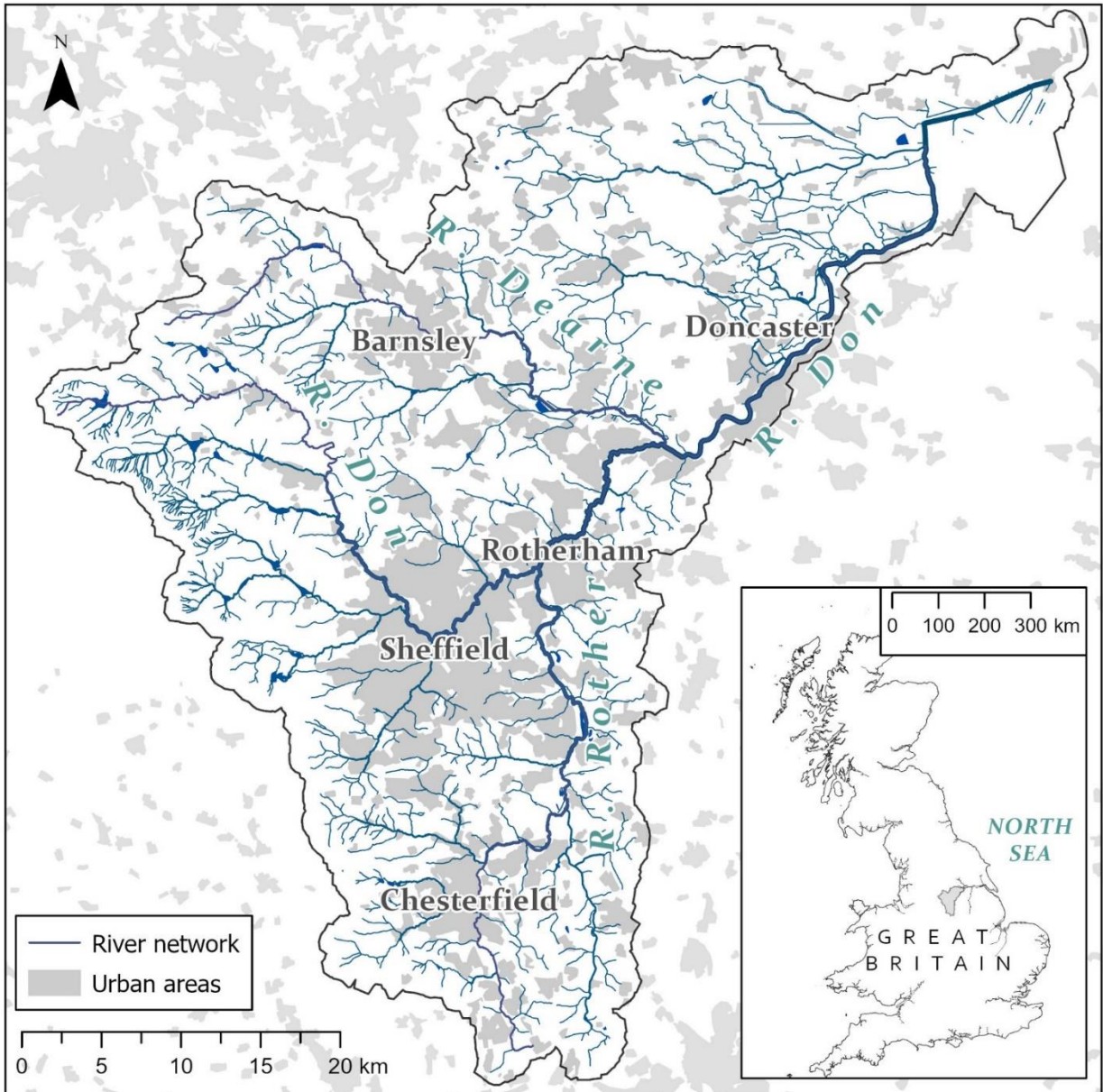

**Figure 1.** The rivers and urban areas of the Don Catchment. Contains OS data © Crown copyright and database right 2021.

While human activity has shaped the river landscape of the Don Catchment for millennia [12], it remained largely rural until the Industrial Revolution in the eighteenth century [13], at which point began an intensification of human activity that has utterly transformed many of the rivers and streams [14]. A few historical sources describe the catchment prior to the Industrial Revolution and provide insight into the rich biological abundance and diversity that existed when it was still in a relatively natural state. For example, the current abundance of European Eel (*Anguilla anguilla* L.) in the catchment is low, and the species is classed by the IUCN as Critically Endangered [15]. However, a vicar in the seventeenth-century described a very different situation:

> "*Commonly every May such vast numbers of young eels comes over the wheels with the waters and run into the mills, that they are forced to give over working and to send into town for the swine to devour them, for they are as innumerable as sand grains on the seashore*". [16]

This is just one of a number of historic records compiled by Firth [14] that consistently indicate that the catchment had a much more abundant and diverse flora and fauna in pre-industrial times.

### 3. Phase 1—Industrialisation, Urbanisation, Ecological Devastation, and Disconnection of Rivers from People

One of the first major human impacts on the river ecosystem was the impoundment of the rivers by weirs, a type of low-head run-of-the-river dam (see Figure 2). Weirs were built to raise water levels upstream so that water could be drawn into side channels (known in the region as goits) that powered water wheels and associated machinery. They were constructed for a variety of purposes, including the milling of corn, processing of wool, and metal working [13]. Water wheels have likely been present in the catchment for at least a thousand years as the Domesday Book makes references to a small number of mills at settlements in the Don Catchment [17], but over the centuries progressively greater numbers were built. Sheffield became renowned for its cutlery industry, which was present in the city at least as far back as the thirteenth century [18]. This and other medieval metal working trades flourished in the west of the catchment due to the availability of fast flowing rivers for waterpower, oak woodlands for charcoal production, iron ore for smelting and grit stone for grinding [19]. Waterpower was harnessed for a variety of purposes including working bellows to blow air into forges, moving hammers, rollers, and cutters to shape metal, and turning grinding wheels to sharpen blades. So important was waterpower in the Sheffield area that by the late eighteenth century almost every suitable reach of river had a water-powered site, with five to six per mile on steep gradient rivers such as the Porter and Rivelin [19]. Weirs were also built for other purposes, and some especially sizeable ones were erected when sections of the Don, Dearne and Rother were made navigable in the eighteenth and nineteenth centuries to facilitate the bulk transport of materials in the catchment.

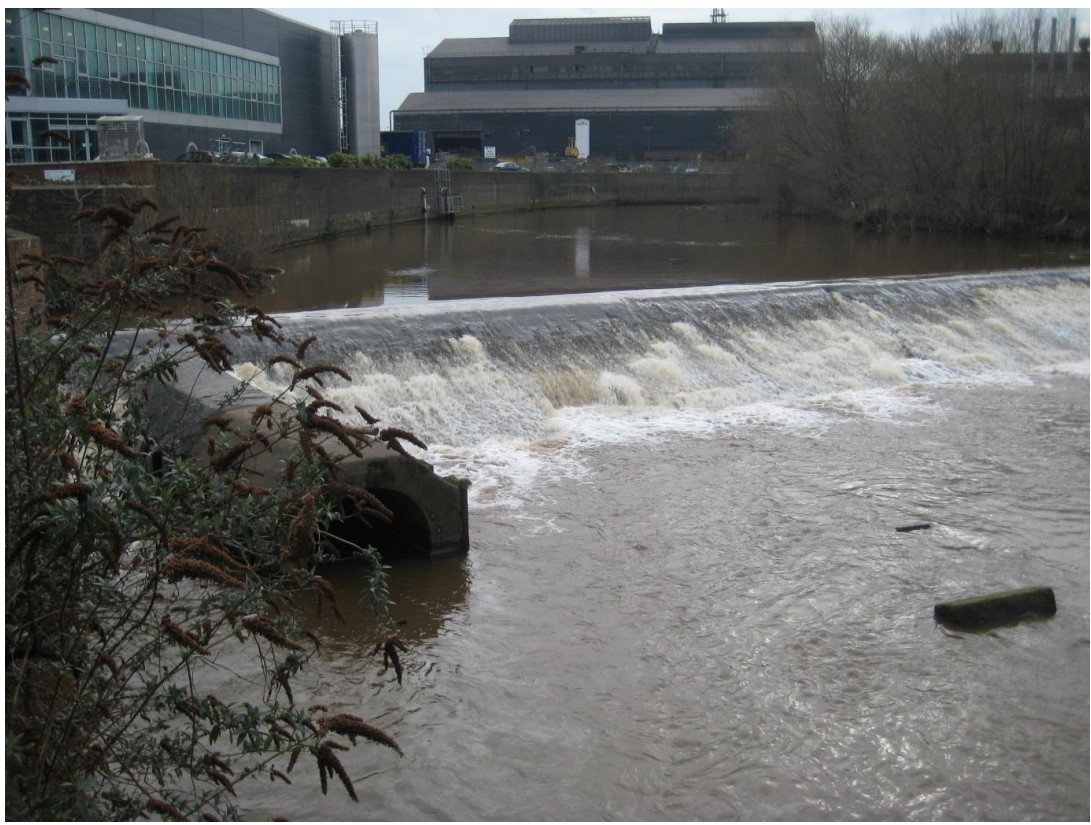

**Figure 2.** Brightside Weir in the River Don in Sheffield. Photograph: DCRT 2009.

Weirs have had a profound impact on river ecosystems [20]. Water levels are increased on the upstream side, creating an impounded slow flowing reach with uniform flow and morphological characteristics, similar to that of a canal. This impoundment effect can extend hundreds of metres upstream of a weir, depending on the gradient of the river. Weirs also represent a barrier to the movement of river biota through the river network, and so reduce or even sever ecological connectivity for certain species [21]. The weirs in the Don Catchment severely fragmented the river network, inhibiting the ability of fish to move effectively throughout the catchment to breed, feed and shelter. This was particularly devasting for migratory fish that need to travel between the sea and freshwater ecosystems and was a leading factor in causing both the extirpation of Atlantic Salmon (*Salmo salar* L.), and a major range contraction in European Eel [14].

The period from the seventeenth to the twentieth century saw industry develop rapidly in the catchment. Sheffield and Rotherham became major steel producing centres due to the abundance of coal in the region and the local discovery of important technological advances such as crucible steel and the Bessemer process [19]. By the mid-nineteenth century Sheffield produced nearly half the European output of steel [18]. Elsewhere in the catchment, Barnsley, Chesterfield and Doncaster also established a wide range of light and heavy industries, including coal mining, engineering, textiles, and glass production [13]. The mining of the South Yorkshire Coalfield was particularly important, and from the mid-nineteenth century onwards large areas of the catchment were given over to collieries and ancillary industries and activities such as coke production and railways [13].

When the catchment's current and former industrial sites are mapped, it is extraordinary how strongly associated it is with the catchment's rivers (see Figure 3). This close association has arisen for several reasons. Firstly, not only did waterpower remain important until the early twentieth century, but early industry often acted as a seed around which later industry grew. Rivers also provided water for industrial processes such as drenching steel, and served as a convenient drain into which effluent could be discharged. Furthermore, river valleys provided flat land that suited the building of large industrial facilities, as well as the construction of canals and railways that were essential for bringing raw materials and taking away products.

The growth in industry attracted large numbers of workers, who migrated to the area, causing a massive expansion of populations. For example, the population of Sheffield grew from 14,531 in 1736 to 135,310 in 1851 and 380,793 in 1901 [18]. The sewage and waste produced by the population, along with the effluent generated by industry, took a severe toll on water quality [14].

Speaking of the Sough Dike in mid-nineteenth century, one Barnsley resident said:

> "*Formerly its filthy and turbid waters flowed uncovered close to the main street, which circumstance gave strangers a very disagreeable impression of the town. Before the water reaches the town it is perfectly clear, but during its further course, dye-houses, calenders, and sewers complete its defilement*". [22]

In roughly the same period Sheffield's rivers were described with even more vivid language:

> "*The three rivers sluggishly flowing through the town are made the conduits of all imaginable filth, and at one particular spot which we shall presently describe, positively run blood. These rivers, that should water Sheffield so pleasantly, are polluted with dirt, dust, dung, and carrion; the embankments are ragged and ruined; here and there overhung with privies; and often the site of ash and offal heaps—most desolate and sickening objects . . . A plank bridge over the Sheaf here shows dead dogs and cats floating on the slimy waters, and a terrible condition of the partially-walled banks, through outlets in which fluents of excremental slush ooze into the river. It is 26 feet wide at this point, and has all the appearance of having once been a vast covered sewer, and having now become ruined, it being impossible to realize that the objects in it were ever intended to meet the sight or smell*". [23]

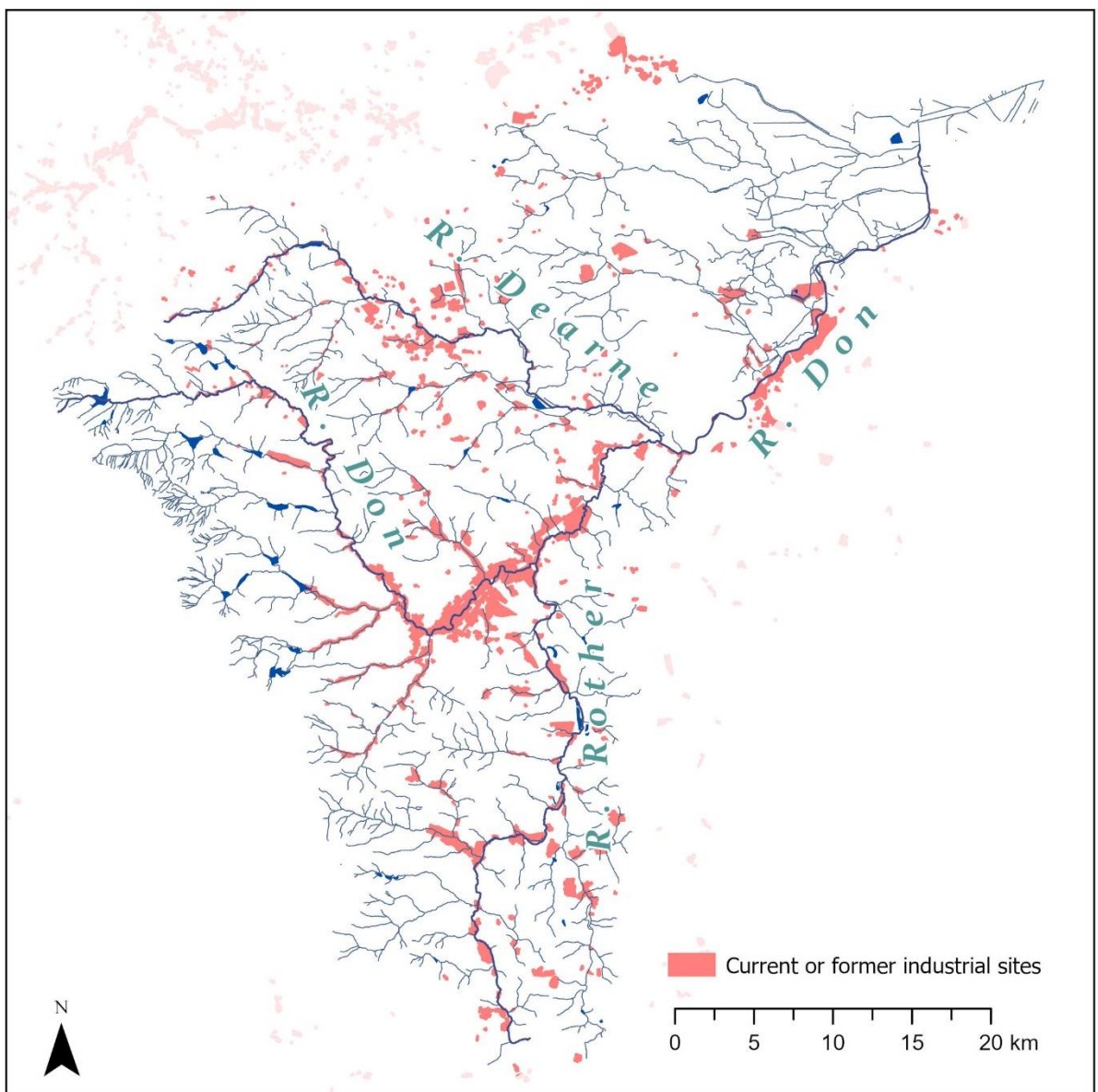

**Figure 3.** Current or former industrial sites in the Don Catchment. Information based upon Historic Environment Characterisation Data © South Yorkshire Archaeology Service 2008 [24], Historic Landscape Characterisation Data © West Yorkshire Archaeological Advisory Service [25] and Derbyshire County Council [26]. Contains OS data © Crown copyright and database right 2021.

Combined sewer systems (ones that combine surface runoff and sewage, in contrast with the modern separate sewer approach which keeps these fluids in different sewer systems) and waste-water treatment works were established, but these fell far short of dealing with the large volumes of effluent produced by the population [14]. The cocktail of domestic, commercial and industrial pollution resulted in the severe pollution of the Don, Dearne, Rother and numerous tributaries, and led to eradication of most forms of aquatic life from much of the catchment's river network [14].

Because of the gross pollution and unpleasant nature of the urban rivers, it is therefore unsurprising that the mills, workshops, steelworks and factories that lined them were orientated away from them, with little consideration of retaining physical or visual access for the public. The resultant effect on the river corridor was to turn them into 'urban canyons' (e.g., Figure 4). In places this process of entombing rivers in the urban fabric was taken a step further by culverting over them, as happened to the aforementioned Sough Dike in Barnsley and the lower reaches of the River Sheaf and Porter Brook in Sheffield

(see Figure 5). Numerous smaller streams were simply buried under development. It is estimated that half the total stream length and over 100 natural springs in the Sheffield area have been lost in this way, with the flow of some incorporated into the sewer network, in a process known as 'stream capture' [27]. As a result of the inhospitable state of the rivers and a riparian 'wall' of industrial buildings, communities became increasingly disconnected from the rivers; they could not easily access them, see them, nor did they want to.

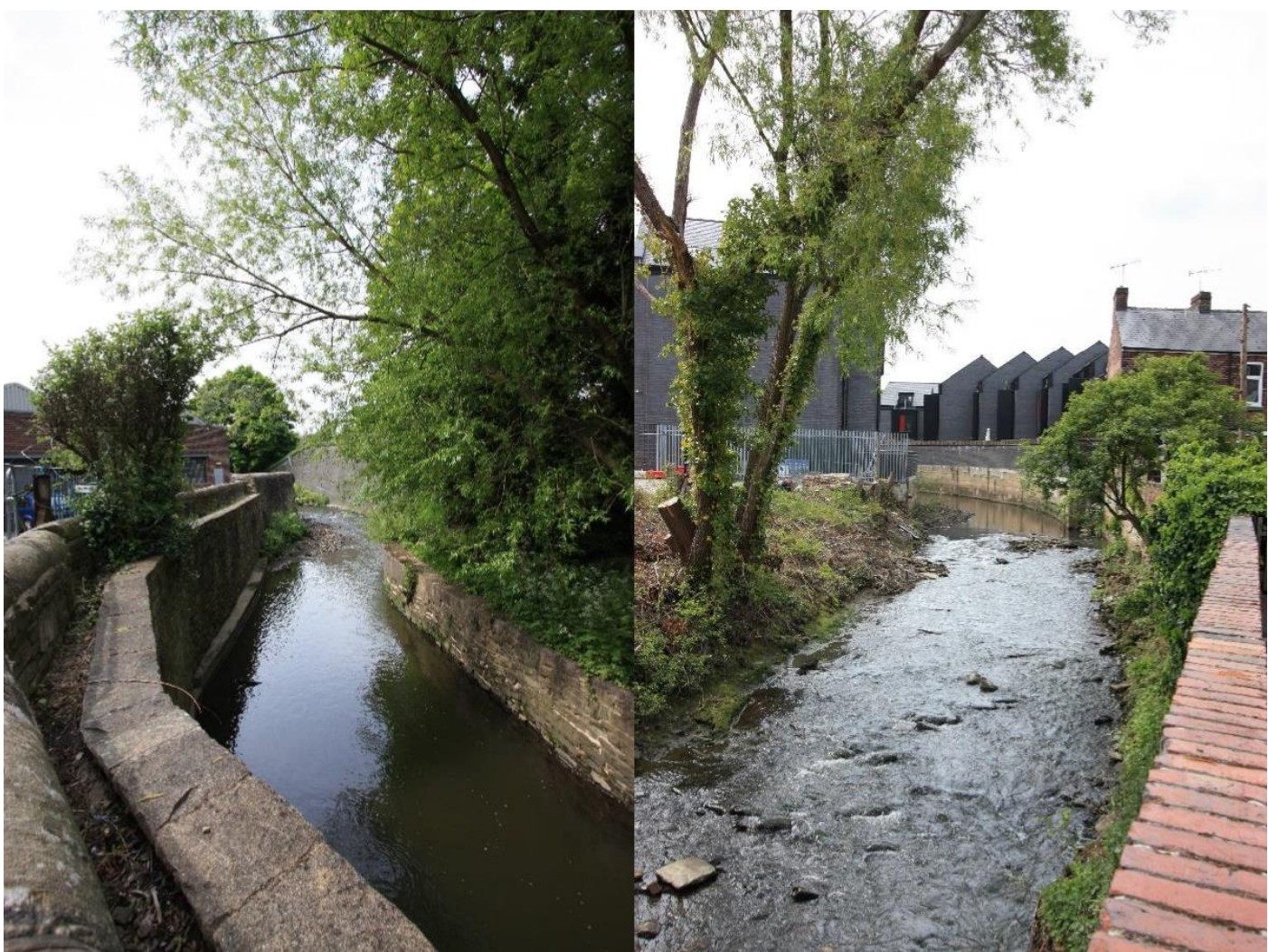

**Figure 4.** The 'urban canyon' effect caused by the building of industry in Sheffield right up to the edge of the River Sheaf, one of the city's main rivers. Photograph: DCRT 2021.

Whilst the majority of the catchment's rivers and streams have not been culverted or buried, they have been highly modified in other ways. They have frequently been straightened to make space for development, especially for railways, which tended to follow valley bottoms. Instead of following their natural meandering course, many rivers and streams now have stretches that run parallel to railways lines (see Figure 6). Further morphological alteration has been driven by a process of channelisation, embankment, and dredging, as rivers are controlled in a quest to prevent erosion, improve land drainage, increase conveyance to reduce flood risk, and in some instances, maintain navigable reaches of river (Figure 7). The lower Don is an extreme example, now 'straightjacketed' in an unnaturally deep narrow uniform channel, disconnected from surrounding lands by high flood embankments (see Figure 8). This sense of containment is heightened in places where the subsidence of adjacent land due to coal mining has resulted in embanked rivers being perched above the surrounding land. Embankments and subsidence mean that pumping

stations are now required to pump many streams and drains in the lower catchment into rivers. The physical modification of the rivers has been devastating for wildlife. The historic mosaic of meandering channels, ponds, reedbeds and water meadows that once existed in valley bottoms, have been almost completely destroyed [14]. Just how extensive physical modification has been in the catchment is made apparent by the Water Framework Directive (2000/60/EC) [28], which classifies most of the river waterbodies to which the legislation applies as 'Heavily Modified Waterbodies' (58 out of 70).

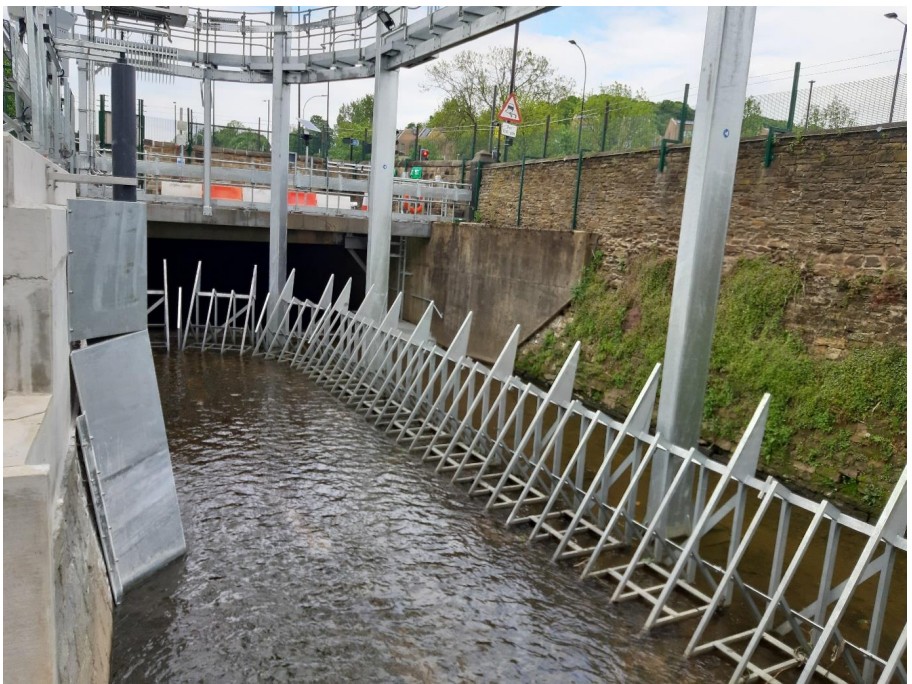

**Figure 5.** The River Sheaf in Sheffield immediately before it disappears under a culvert. Photograph: DCRT 2021.

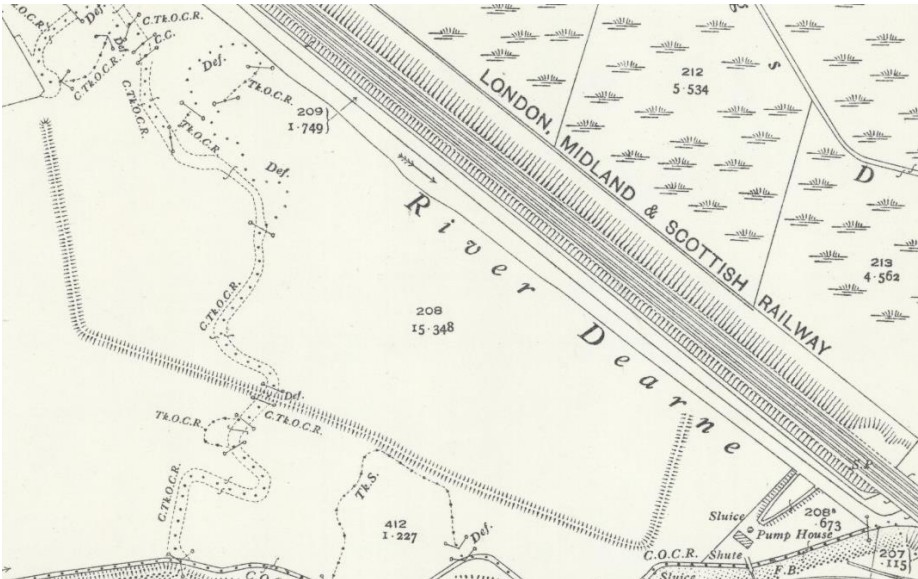

**Figure 6.** A historic Ordnance Survey map dating from around 1930 showing a reach of the River Dearne straightened so that it runs alongside a railway line [29].

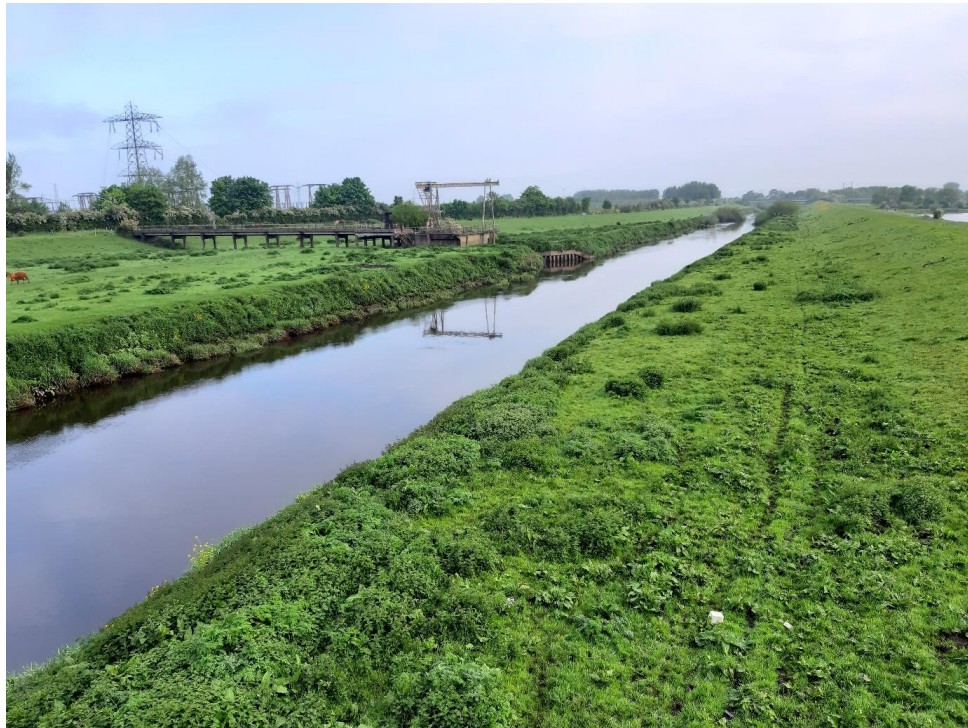

**Figure 7.** Examples of heavy river channel modification in Sheffield. Photograph: DCRT 2021.

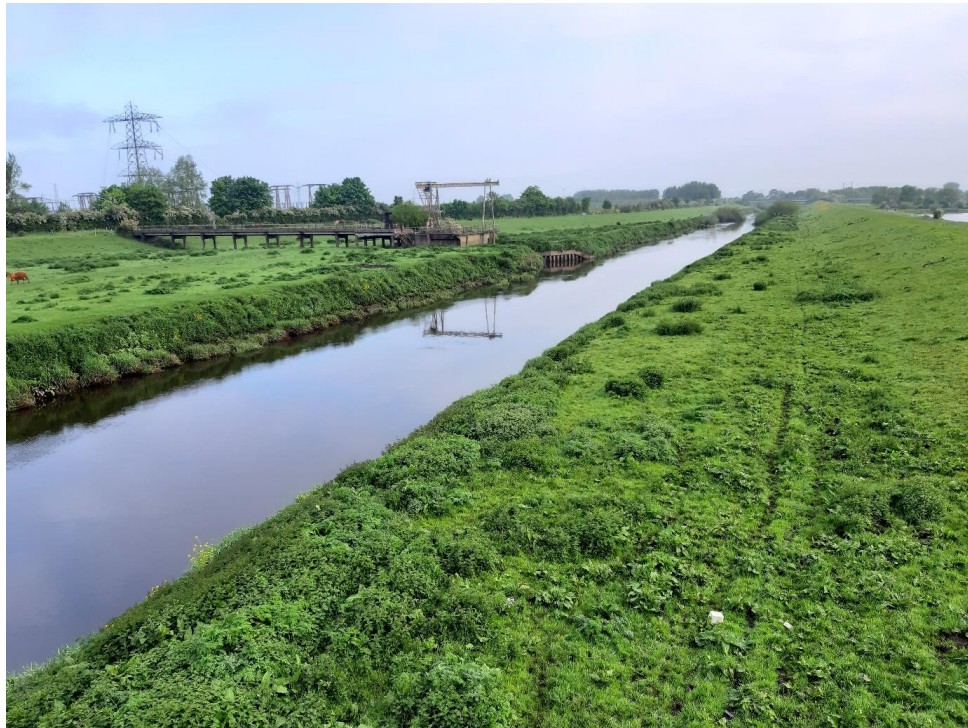

**Figure 8.** A straightened, deepened, and embanked reach of the lower River Don. Photograph: DCRT 2021.

## 4. Phase 2—Deindustrialisation, Partial Ecological Recovery

In the 1970s and 1980s a process of massive deindustrialisation occurred in the catchment, with major declines in coal mining, steel and associated industries [30]. This resulted in a corresponding decline in pollution from effluent [14]. In addition, better management and treatment of domestic wastewater, such as the construction of Sheffield's interceptor sewer in the 1980s and 1990s, provided major additional water quality gains [14]. As a consequence of improved water quality there was a marked improvement in the condition of most of the catchment's river ecosystems [12,14,31]. Deliberate reintroductions of species such as Grayling (*Thymallus thymallus* L.), Common Chub (*Squalius cephalus* L.), Common Dace (*Leuciscus leuciscus* L.), Common Barbel (*Barbus barbus* L.) and Water Crowfoot (*Ranunculus* sp.) were successful [14]. Other species such as Kingfisher (*Alcedo atthis* L.), Grey Heron (*Ardea cinerea* L.) and Otter (*Lutra lutra* L.) spontaneously recolonised parts of the catchment from where they had long been absent [12]. Many non-native species also became established, thrived and have become major components of the river ecosystems. Sheffield's nascent river ecosystems have been described as 'recombinant', a fusion of native and exotic species [12]. Example non-native species can be drawn from a wide range of taxonomic groups, geographic origins, and trophic levels, such as the Common Fig (*Ficus carica* L.), Himalayan Balsam (*Impatiens glandulifera* Royle), Buddleia (*Buddleia davidii* Franch.), Jenkins Spire Snail (*Potamopyrgus antipodarum* Gray), Demon Shrimp (*Dikerogammarus haemobaphes* Eichwald), Signal Crayfish (*Pacifastacus leniusculus* Dana), and American Mink (*Neovison vison* Shreber).

The decline of industry has had a major impact on the urban river landscape, but the diverse patterns of industrial development in the catchment has imbued different reaches of river with their own distinct character. This is apparent not only between settlements, but also within them, as illustrated by the examples of Sheffield and Barnsley. The predominant industrial activity associated with the high-gradient fast-flowing rivers in western Sheffield was small-scale and water-powered (see Figures 9 and 10). When this technology became obsolete, these water-powered sites were abandoned and became derelict, so that today ruins of stone buildings, weirs, goits, earthworks and mill ponds known as dams punctuate valley bottoms (see Figure 11). The amenity and welfare value of these relatively rural valleys was recognised as far back as the early twentieth century, and philanthropists and public-minded bodies acted to purchase and protect them as public green space, as all the while urban Sheffield expanded on the hills between the valleys [12]. The end result of this benevolent and forward-thinking action is that these river valleys now represent long corridors of green space that run deep into the city.

Further downstream, in and around central Sheffield, the character of the rivers is very different. In places the brick workshops and small factories used by the light metal trades still line the banks (see Figures 9 and 12). Large numbers of these buildings, however, were demolished in the early to mid-twentieth century due to a combination of modernisation and clearance policies and inner-city road schemes [13]. On the footprints of the old buildings, offices, 'city living' and student accommodation, and civic buildings have been built, and some plots have been left open as car parks or brownfield land [13,19]. This process has created a diverse mix of urban form in the river corridors; old and new, crowded and open, river hugging, set back, and 'river indifferent'.

Further downstream still, in the Don Valley below Sheffield, the river landscape has yet a different character. The enormous steelworks that once filled the valley largely closed following the collapse of the steel industry in the late 1970s [19]. The steel industry has not entirely disappeared, and some large corrugated-steel clad sheds remain, but most of the steelworks have been demolished. Those that were adjacent to the Don have been replaced by large warehouses, sheds and compounds associated with light industry, retail parks, a large shopping centre, and brownfield sites in varying states of natural regeneration [13]. While there is a strong industrial and post-industrial character to the river landscape, the immediate riparian zone of the Don has a surprisingly wild urban hinterland character (see Figure 13).

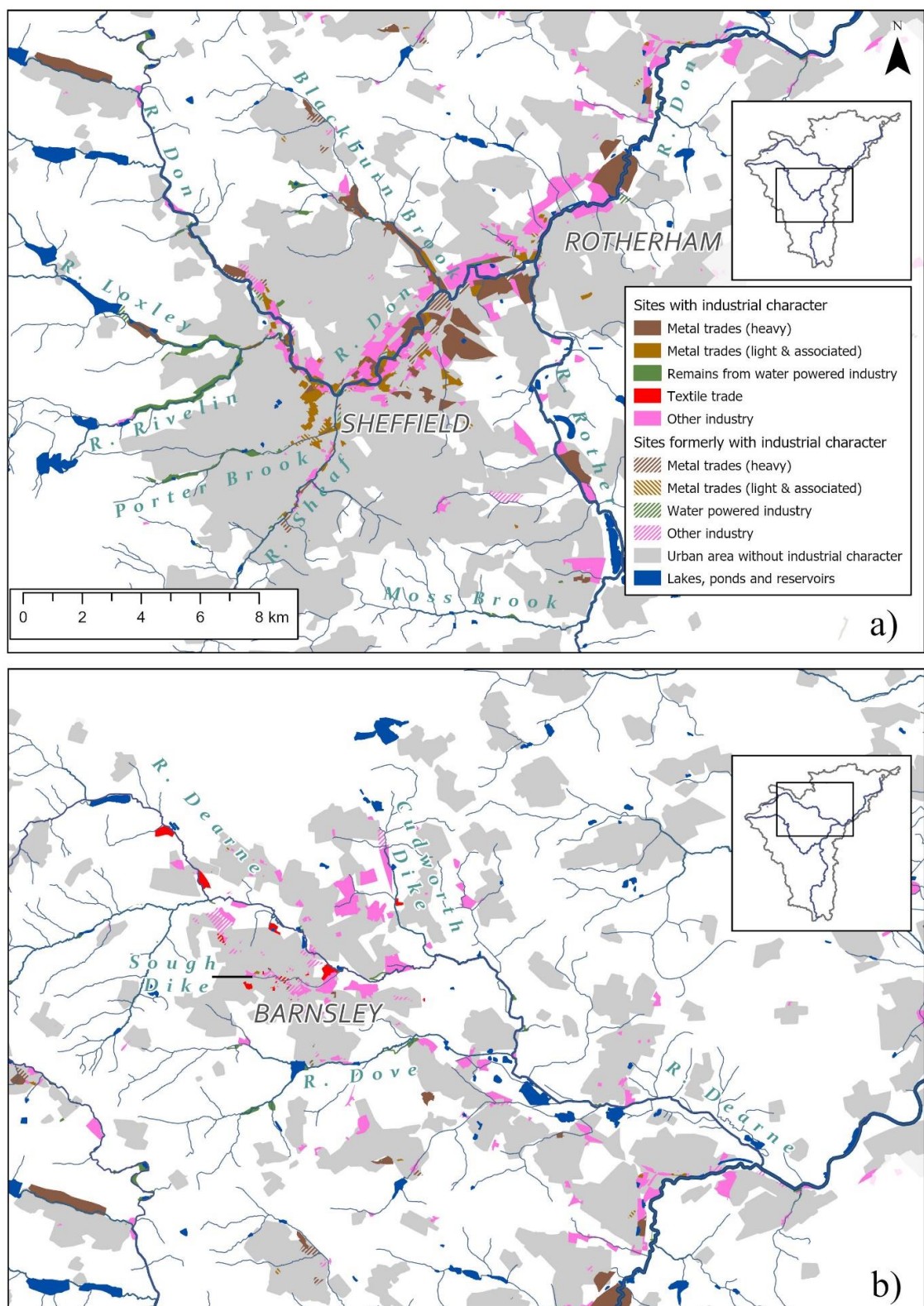

**Figure 9.** The current or former industrial character of (**a**) sites in the Sheffield and Rotherham area and (**b**) sites in the Barnsley area. Information based upon Historic Environment Characterisation Data © South Yorkshire Archaeology Service 2008 [24]. Contains OS data © Crown copyright and database right 2021.

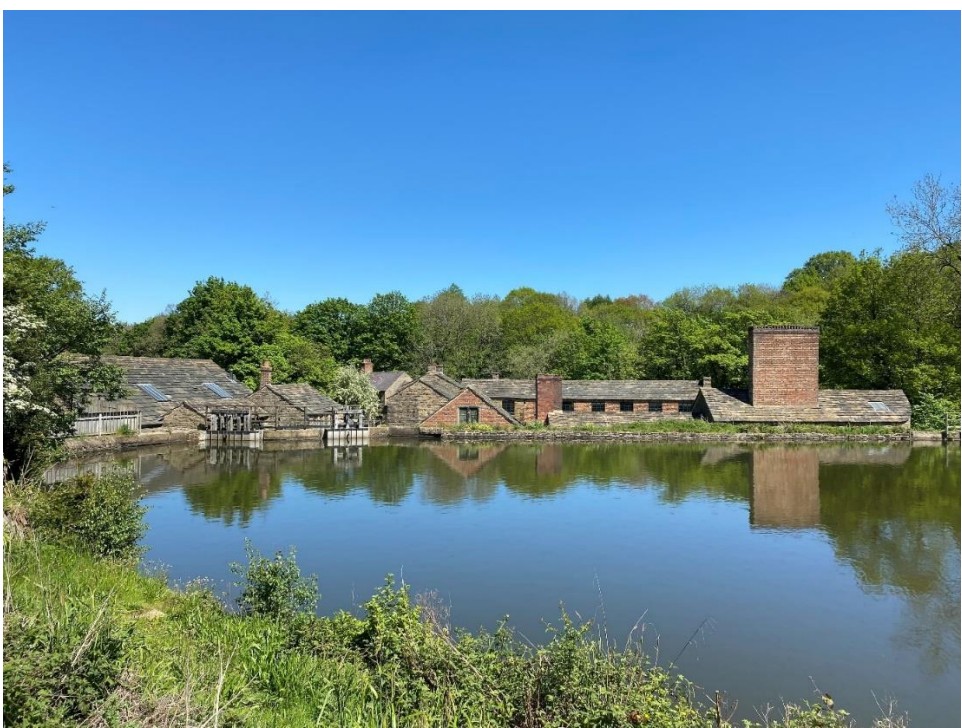

**Figure 10.** Abbeydale Industrial Hamlet, an unusually well preserved water-powered site on the River Sheaf that was an active metal working site for over 500 years. Photograph: DCRT 2021.

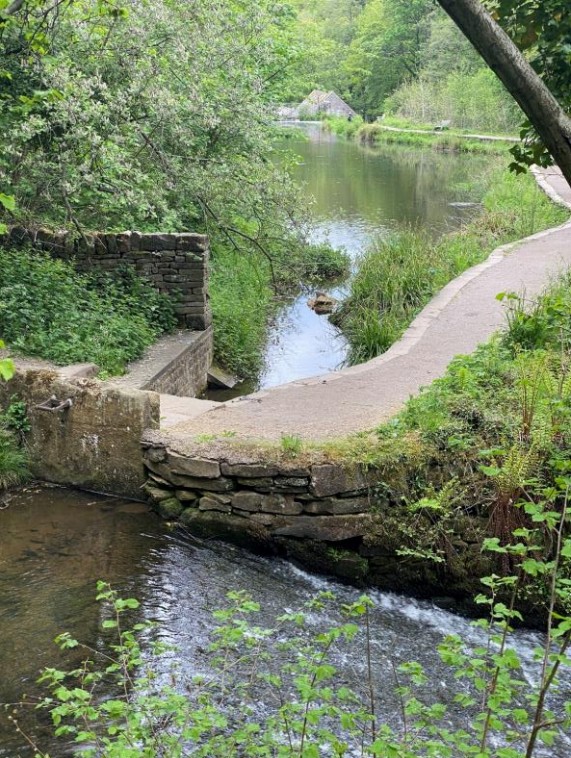

**Figure 11.** A mill pond ('dam') built for a former water-powered site. Photograph: DCRT 2021.

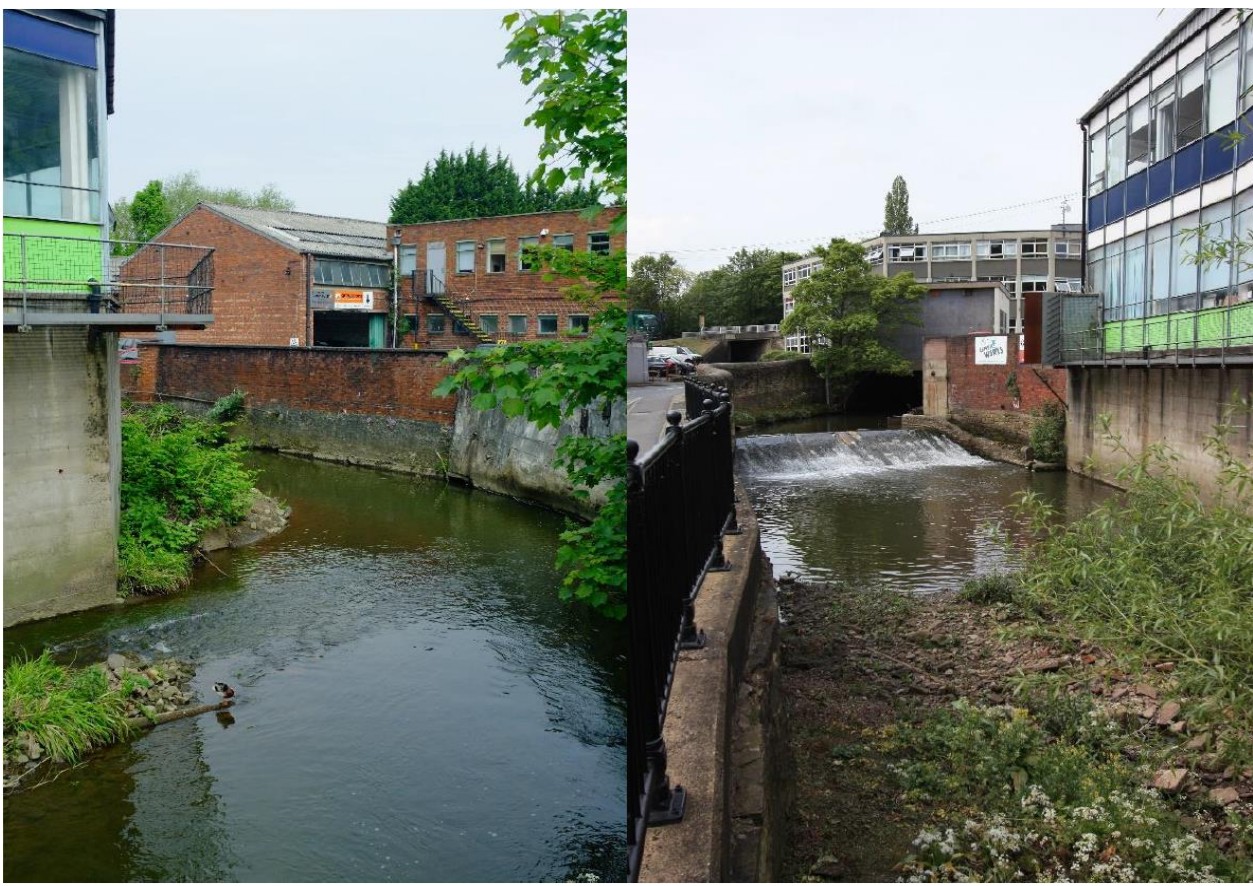

**Figure 12.** Former or current light industrial buildings on the banks of the River Sheaf in Sheffield. Photograph: DCRT 2021.

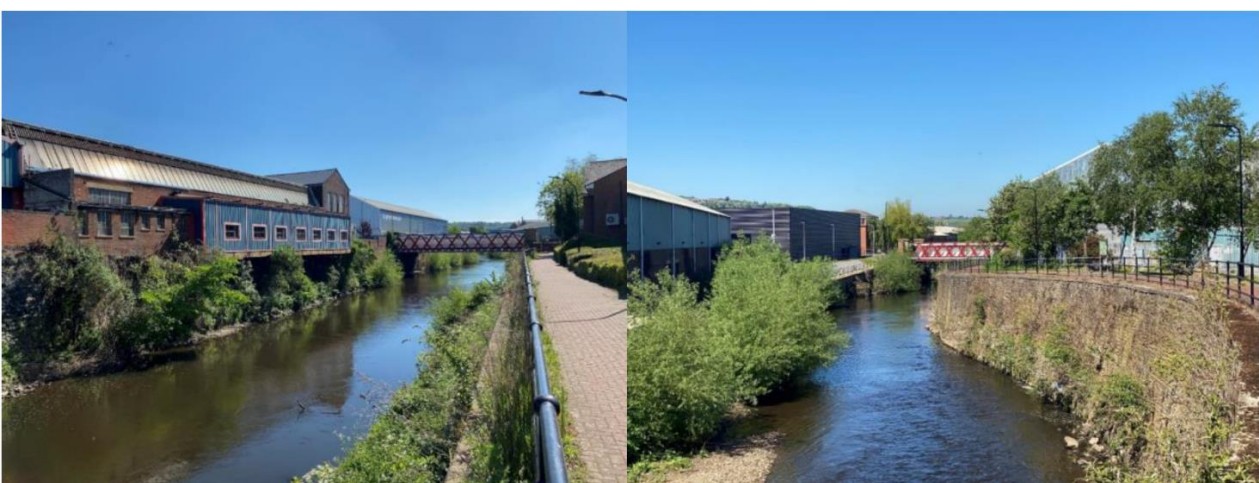

**Figure 13.** Mix of old and modern industrial buildings line the River Don downstream of Sheffield. Photograph: DCRT 2021.

The effect of industrialisation on Barnsley's rivers offers a contrast to that of Sheffield. The town originally developed on the Sough Dike [22], and despite being a relatively small watercourse, industry has concentrated along its length (see Figure 9). The metal industry has been far less important in Barnsley, and so there are far fewer water-powered sites, and the steel and iron industry did not dominate the town like it did in Sheffield. Other industries assumed a more important role, including the production of linen and coal mining [13]. This did not fill the Dearne Valley to the extent that the steel industry filled the Don Valley below Sheffield. Nevertheless, coal mining had a major impact on the

landscape of the Dearne Valley, with collieries, a network of railways to serve them, and huge mounds of spoil, creating a landscape of mineral extraction. In addition, subsidence due to the mining activity has caused a series of small lakes to form in the valley bottom.

The industry that was once present in the Dearne Valley has largely disappeared, with the dismantling of the railways and collieries, as well as the landscaping of large areas of wasteland. This has not been replaced by substantial amounts of new urban development, and so the valley is now a large continuous green space that runs through the wider urban Barnsley area. This has allowed the establishment of the Dearne Valley Park in 1980, and for other large areas to be set aside as public parkland, nature reserves and remediated agricultural land [13]. The value of the open water, wetland and naturally regenerating spoil heaps as important habitat for wildlife has been recognised by Natural England (a non-departmental public body with nature conservation responsibilities) and has resulted in the notification of the Dearne Valley Wetlands Site of Special Scientific Interest. Despite this, in many places the mounds, earthworks and detritus of industry are still apparent, creating a 'downgraded', neglected, and post-industrial character [13]. As is the case with the Don, the history of pollution and degradation of the Dearne river corridor has resulted in the persistence of negative attitudes towards the river.

## 5. Improving the Don Catchment's Urban Rivers for the Environment and for People

The ecological recovery and dereliction of riverside land caused by deindustrialisation has provided civil society, local authorities, the regional water company, and other public-minded organisations with an opportunity to improve the rivers for social, cultural, environmental and economic good [32]. Aspirations have grown over the last four decades, with visions and strategies presented in many documents including the latest Catchment Plan for the Don, Dearne and Rother Catchment 2021–2026 [33], which sets out a vision of 'Healthy, Resilient Rivers for Nature and People' and details progress that has been made towards this aim.

Numerous organisations and partnerships are delivering interventions to improve the catchment's urban rivers, including amongst other things, deculverting projects [34], Sustainable Drainage Systems, riparian 'pocket parks' and riverside nature reserves [33]. However, here we discuss interventions and initiatives that the Don Catchment Rivers Trust has been directly involved in.

### 5.1. Restoration of Ecological Connectivity through the River Network

Ecological connectivity through the river network of the Don Catchment remains highly fragmented due to the presence of weirs, reservoirs, culverts, outfalls, sluices and pumping stations. Of these, the catchment's >200 weirs were thought to be inhibiting the return of highly migratory fish such as Atlantic Salmon and Sea Trout (*Salmo trutta* morpha *trutta*), European Eel, and Sea Lamprey (*Petromyzon marinus* L.), all of which must travel between marine and freshwater ecosystems to complete their lifecycles.

There has been a long-standing ambition to see Atlantic Salmon recolonise the catchment, and one of the original motivations behind the establishment of the Don Catchment Rivers Trust was to create an organisation that could take action to help achieve this objective. Salmon hatch from eggs laid in rivers and streams, but mature in the sea, before homing back to their natal rivers to reproduce. They have the capacity to colonise new rivers, as their homing ability is imperfect, and a small fraction of a population end up migrating up and reproducing in a different river to which they were born [35,36]. This 'straying' has been apparent in the Don Catchment for decades, where despite no salmon reproducing in the catchment, small numbers of adults in the North Sea still entered the Don. These individuals attempted to migrate upstream, but were halted by the first barrier they encountered, far downstream of suitable spawning habitat.

The impact of weirs as ecological barriers can be mitigated by either removing or breaching the structure, or by installing a fish pass. Efforts to address barriers on the Don began in 2001, when a bypass channel containing a rock ramp was built by the UK

Environment Agency to enabled fish to circumvent Crimpsall Sluice, just upstream of Doncaster (see Figure 14). Since then, many more fish passes have been built by a number of organisations.

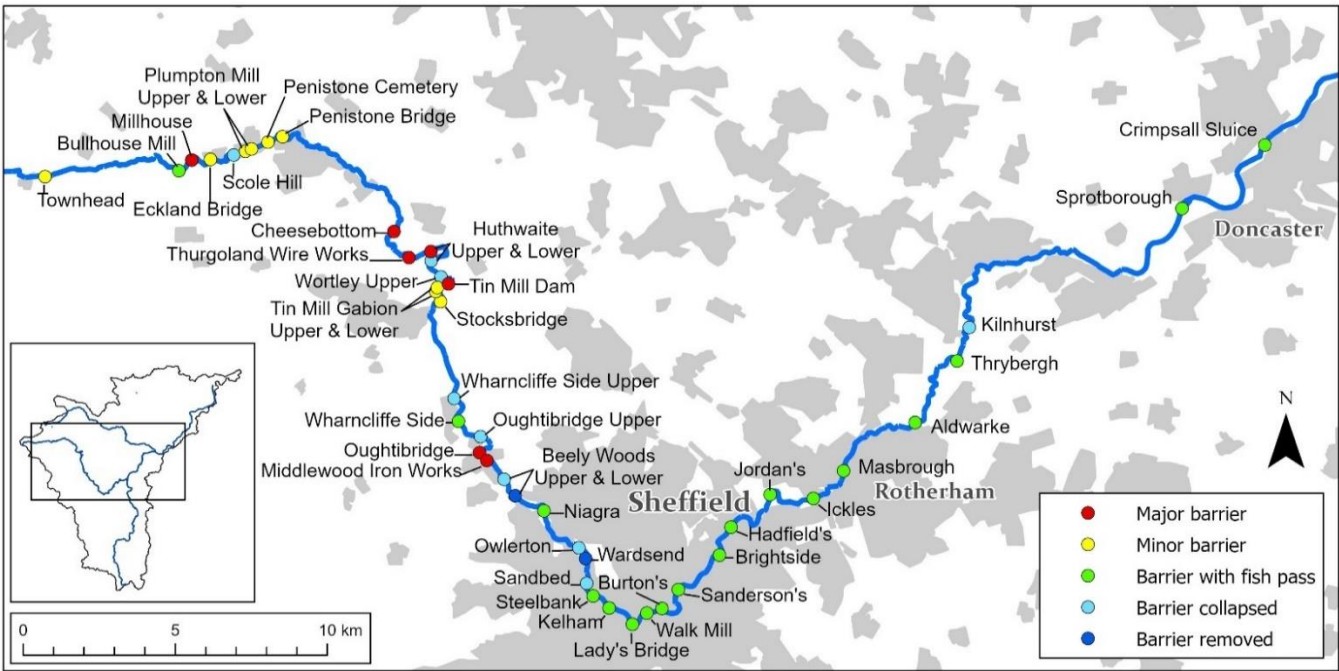

**Figure 14.** Status of barriers on the River Don to the upstream migration of salmonids. Contains OS data © Crown copyright and database right 2021.

From an ecological perspective, the removal of weirs is usually a more desirable intervention than the installation of fish passes [37]. Not only does this reinstate natural processes such as sediment transfer and restore more natural hydromorphological conditions upstream, but it also avoids the need for a fish pass. While weir removal will usually fully restore ecological connectivity, fish passes often only partially restore connectivity, with the outcome dependent on their type and context [38,39]. Some fish species may not have the swimming capacity or exhibit the behaviours necessary to successfully ascend a fish pass [40]. And even species that have the swimming ability to ascend a fish pass may well still be delayed or even completely fail to ascend a pass depending on how effectively they find the pass entrance [40]. Weir removal has often not been possible in the Don Catchment as most of the weirs of the lower Don have the important function of maintaining water levels that keep the river and associated canal cuts navigable, while some weirs higher in the catchment have significant heritage value due to their important role in the development of the region's industry. However, it is hoped that it will be possible to remove more weirs in future years.

In 2020, a fish pass was built on Masbrough Weir in Rotherham, which was the last in a series of fish passes between the sea and Sheffield (see Figure 15). The city is believed to be the first location on the Don where salmon migrating upstream would encounter suitable spawning habitat (though larger amounts of better habitat is believed to occur upstream of Sheffield). Early indications suggest that the chain of fish passes is beginning to work. Two adult salmon were recorded in Sheffield in January 2019, the first records of salmon in the city for over 200 years. In 2020 a salmon parr (juvenile salmon) was caught by an angler in Sheffield, and there were four sightings of adult salmon (one confirmed by a UK Environment Agency expert, three sightings suspected to be salmon).

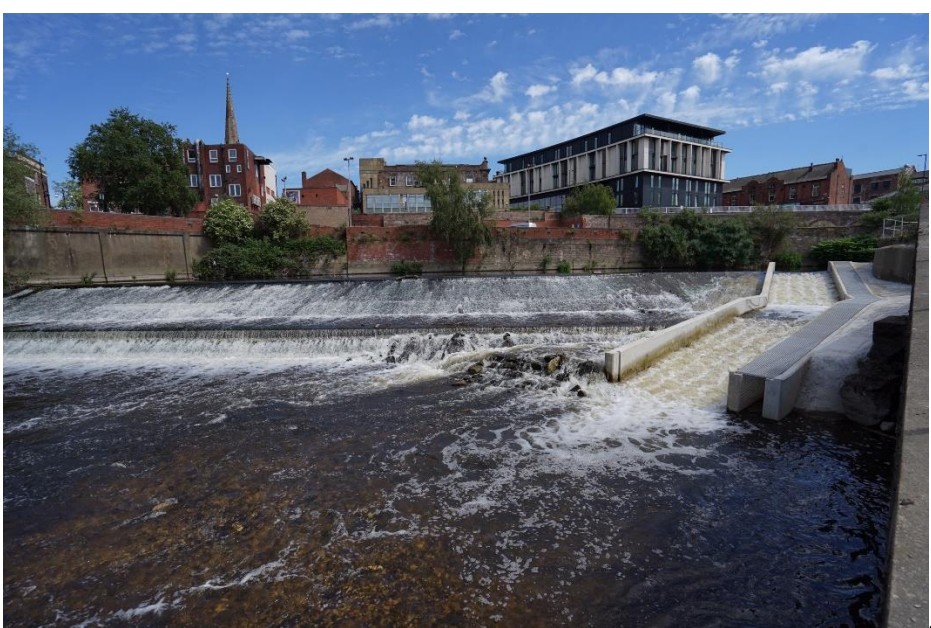

**Figure 15.** A Larinier fish pass built 2020 on Masbrough Weir in the River Don at Rotherham. Photograph: DCRT 2020.

*5.2. Natural Flood Management*

Flood risk is at the forefront of many people's minds when it comes to urban rivers. This is very true of the urban centres of the Don Catchment, which were badly hit by flooding in 2007 and in 2019, when on both occasions over 1000 homes and a similar number of businesses were affected [41,42]. Though climate change is thought to be exacerbating flooding in the UK [43], the legacy of river modification and land use change in the Don Catchment are also significant contributing factors [44]. The loss of natural habitats such as woodlands and wetlands, replaced by intensive agriculture, and increasing areas of impervious surfaces has led to higher runoff rates of stormwater and larger flow peaks in the river network. The river modifications outlined earlier, such as the straightening, channelisation, and disconnection from floodplains, serve only to speed storm flows up further [44]. As a result, the flows of modern urban rivers have become more flashy, and despite extensive flood defences, many communities are vulnerable to flooding. An additional concern is the aforementioned combined sewer system that combines both surface runoff and sewage. Such outdated systems are more prone to overloading with surface runoff than modern day separate sewers, and cause pollution when sewage spills from Combined Sewer Overflows (CSOs) into the rivers. While this is only meant to occur occasionally during heavy rainfall events, data collected for 2019 by the regional water company shows that many discharged much more continuously, with one discharging 40% of the time over the year [45].

Traditional methods of reducing flood risk along urban rivers have generally taken the form of hard engineered structures such as flood walls and gates, but there has been a shift in the last decade to a view that recognises the value of combining both hard and soft engineering approaches. Soft engineered approaches, commonly referred to as Nature-based Solutions and Natural Flood Management, work by adding nature-like features and processes to the landscape to hold back and slow storm water runoff throughout the landscape, generally upstream of urban centres. When done in the right locations and over wide enough areas, Natural Flood Management can help to reduce both the maximum height of flood waters and the time it takes for those flood waters to reach populated areas downstream, allowing more time to prepare [46]. Interventions include a variety of activities such as moorland restoration, catchment and riparian woodland planting, soil management to increase infiltration capacity, wetland creation, reconnection of rivers to their floodplains (where it is safe to do so), installing barriers to flow pathways, and

the creation of runoff storage areas. In urban areas, Sustainable Urban Drainage systems (SUDs) provide a similar function. Culverts, which are often flow bottlenecks, are being removed where possible. In the Don Catchment, the Trust and several other organisations are now delivering Natural Flood Management upstream of Chesterfield and Sheffield (e.g., Figure 16). One of the main attractions of these measures is their great potential for the co-delivery of multiple benefits, including creating wildlife habitat, water quality improvements, carbon sequestration, and the boost these green and blue spaces can have on wellbeing [47,48]. Some measures can also be installed with the help of volunteers, which provides a means of engaging with local communities.

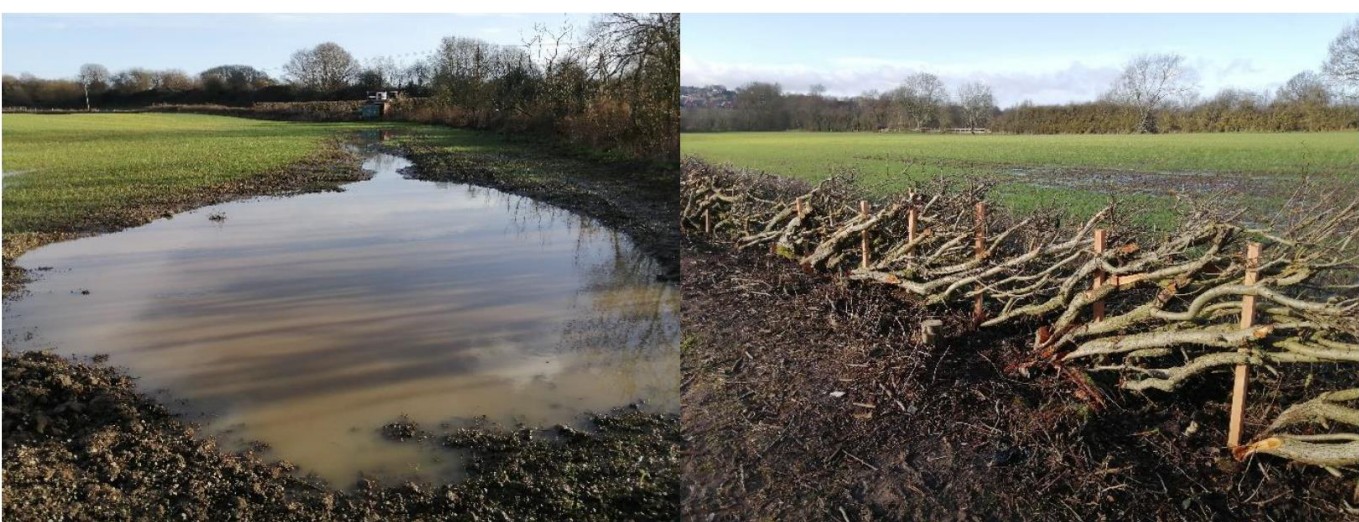

**Figure 16.** Examples of Natural Flood Management interventions created upstream of Chesterfield. **Left**: scrapes dug to intercept and store runoff. **Right**: Hedgerow restored to slow down the flow of runoff. Photograph: DCRT 2020.

### 5.3. Reconnecting the Public with Rivers

The fish passes and other interventions delivered by the Don Catchment Rivers Trust and other organisations are only half of the story of tackling urban river syndrome in the catchment. The Don runs through the settlements of Doncaster, Rotherham and Sheffield, yet when the Trust first started its community engagement programme in these areas in 2015, we found that there was a social and emotional disconnect between people and the river. There seemed almost to be a fear of the Don; considered by many as somewhere where people should not go or needed permission to visit. This is understandable when you consider that around eight generations of people have lived alongside a grossly polluted Don. It was hard for people to think of the river as a pleasant place to visit, or to believe that an ecological recovery was possible, or even that fish could live in urban rivers.

So why should we engage with communities during the improvement of urban rivers? Why not just accept that for some communities, rivers are not their priority? The motivations behind our community engagement programme are multifaceted and interconnected but are ultimately underpinned by a belief that there is a synergy between ecological recovery and community engagement, resulting in a win-win for both rivers and people.

While landscape design and architecture can create accessible, wildlife-rich and aesthetically pleasing urban river corridors, it cannot be assumed that these qualities alone will lead to people valuing and using them. Indeed, we would argue that there are cases of beautifully restored urban river spaces which, because of a lack of a sense community ownership and engagement, are not that successful in bringing people back to the river. Therefore, if we want the public to benefit from the improvement of the urban river corridors then we need people to realise they exist, care about them, and want to spend time there.

An additional motivation is addressing social disadvantage. There are many deprived communities in close proximity to the urban rivers in the Don Catchment (Figure 17). The rapid deindustrialisation of the 1970s and 1980s resulted in a big jump in unemployment in the catchment, with, for example, Sheffield's unemployment rate, which stood at 4% in 1978, soaring to 13.7% by 1991 [30]. Even though there has been significant economic recovery since then, the collapse of the region's industry has left entrenched areas of deprivation in the catchment [49].

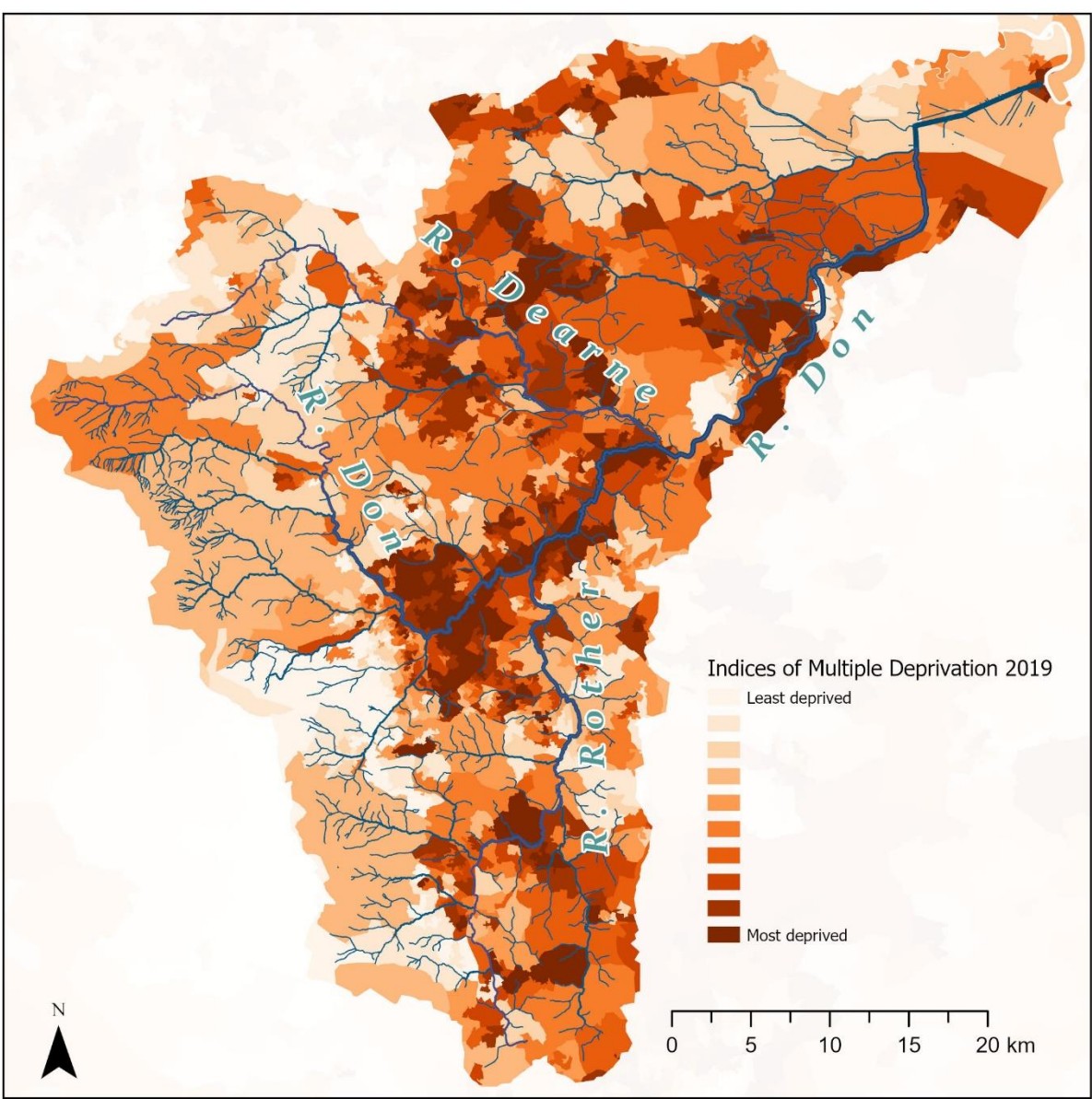

**Figure 17.** The English Indices of Deprivation 2019 [50] in the Don Catchment. Contains OS data © Crown copyright and database right 2021.

Deprived communities have relatively high rates of mental and physical poor health compared to less disadvantaged communities [51]. Urban river corridors therefore offer an opportunity to make a positive difference due to their potential role as public green space and green infrastructure, as this can provide a suite of benefits including improved mental and physical health that can reduce health inequalities, and encourage social interaction, exercise, play and contact with nature [48,52].

Reconnecting people with urban rivers, then, is fundamental to unlocking their potential as assets to local communities. To this end, we have sought to understand the

reasons and drivers of negative perceptions of the rivers, and we have taken the view that with project development it is important not to start with the river and work out towards the community, but to understand the community and place the river within the context of 'urban nature' and 'urban green-blue space'. Our activities have been as much about community engagement as they have about ecological restoration.

One very important aspect to understanding communities and their sense of identity are their cultural ties to industry. For example, Sheffield is commonly known as 'the Steel City', while the nickname of Sheffield United, a major football team, is 'the Blades', in reference to the cutlery industry. Often, the links between sense of identity and industry are intangible; centred around where and how people and their families grew up, and the skills and traditions that they acquired. Historic industrial structures in the landscape can provide a focus for this intangible heritage.

In many instances, the celebration and preservation of industrial heritage in river corridors is complementary with efforts to improve the ecological condition of the rivers, but there is a tension in the case of weirs. These structures are a tangible embodiment of the catchment's industrial heritage, and while many mill buildings and steelworks have long ago disappeared from the urban landscape, weirs are one of the most substantial remaining industrial heritage features. A number of well preserved, visually striking and historic weirs are afforded protection by being listed, or otherwise recognised for their heritage value. For example, the weirs and other remains of water-powered sites in the Loxley Valley have been described as '*far more than local or regional significance: their importance is European, if not more widely international*' [53].

This heritage value attached to weirs has resulted in a divergence of interests between ecological restoration and heritage conservation. As described earlier, from an ecological perspective it is often better to remove weirs to fully reinstate ecological connectivity and a natural river morphology, whereas, from a heritage perspective it is better that the weirs are left unmodified. A compromise has been reached on the Don, with the use of fish passes on weirs with high heritage value, though there are concerns that this necessitates a long series of fish passes, which may be less effective at restoring longitudinal connectivity in river networks compared to weir removal [54].

In 2014 the Don Catchment Rivers Trust began to develop a Heritage Lottery Fund project called the Living Heritage of the River Don that would build two Larinier and three easement-type fish passes on weirs in the Don in Sheffield as well as deliver a major community engagement programme. From the outset, the Trust aimed for the project to both improve riverine ecological connectivity, and to reconnect communities back to the river. Initial pre-project interviews with the public revealed that there were many negative perceptions of the river (e.g., see Figure 18) and identified that there were two main barriers to community participation in the wider project. Firstly, there was the industrial heritage value that some people placed on the weirs. This was something that we did not want to challenge as we too think that the industrial heritage of the river is important. In response to this concern, we raised awareness of the ecological recovery of the Don, and framed the installation of fish passes as a new phase in the history of the weirs, rather than as a destructive process.

The second barrier was disbelief or mistrust that ecological improvement was occurring, and that it was possible for salmon to return to the Don. Although some people had heard stories of salmon jumping below weirs lower down in the catchment, and place names such as Salmon Pastures hint of the earlier existence of a salmon population, some people found it hard to believe that a breeding population of salmon could become reestablished in the catchment. On hearing this project objective people would smile wryly or offer a sceptical 'good luck'.

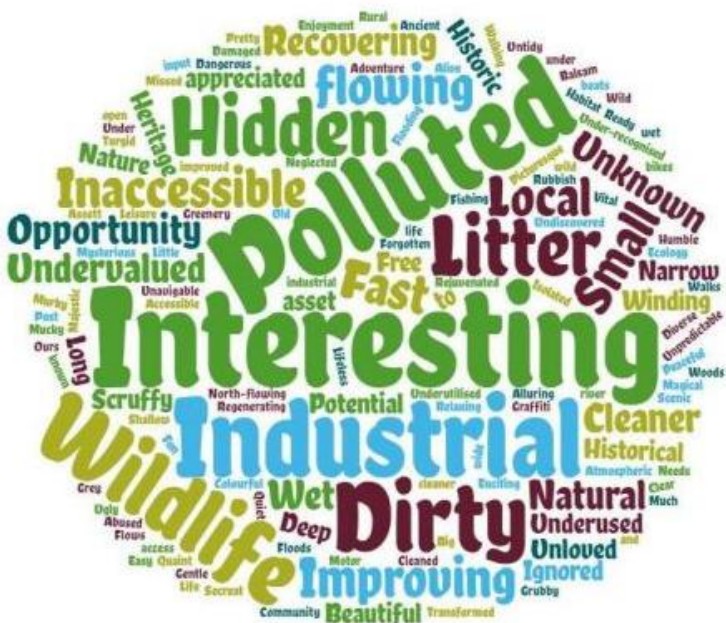

**Figure 18.** Word cloud created from words used to describe the River Don in a community survey conducted in 2015.

Yet along with this negativity it was also apparent that people understood the potential of the urban rivers as assets to local communities, as shown by the following quotes about the River Don from a focus group in 2014:

> " . . . *rivers make cities and it is a marvellous facility waiting to be developed*"

> "*I think [the river] is a little gem for our town and it needs to be brought out to flourish and become a focal point of the town and the wider region*".

The pre-project feedback from local communities led to us deciding to pursue project messaging that promoted awareness of the return of salmon back to the Don, as we felt this would be a powerful way of forging a cultural change in attitudes within our riverside communities. Salmon have the potential to capture the public imagination as they are charismatic and iconic species that many associate with pristine wild rivers, so their return demonstrates that there has been genuine improvement in the catchment's river ecosystems.

In order to reconnect people to rivers the following project aims were also set:

- To inspire and mobilise the community

Community surveys revealed that people wanted river corridors that provided higher quality green space, but they did not feel able to make a difference as individuals. By developing volunteering programmes and river-based activities, such as removing invasive plant species, and clearing up rubbish, we were able to mobilise and empower people, and provide them with a sense that they were directly contributing to improving their rivers (e.g., see Figure 19).

- To foster understanding and change perceptions

For many people the river still felt 'industrial' and severely polluted. To change this view we used a variety of means to engage and educate people. For example, by running school sessions, activity sessions, walks and talks, attending shows and events with a stall, and by establishing a walking route along the Don called the Don Valley Way.

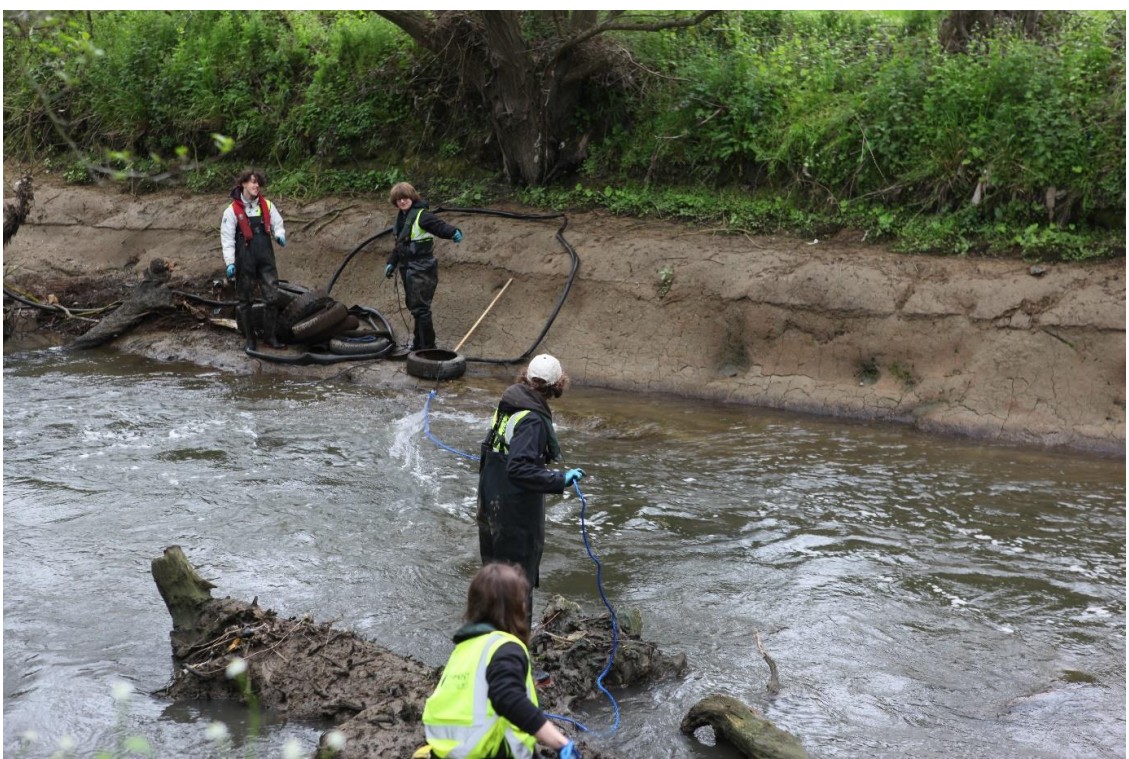

**Figure 19.** Staff and volunteers remove tyres and other rubbish from the River Rother. Photograph: DCRT 2021.

Art also played a major role in the project. In common with experiences reported in the literature [55,56], the Trust found that art offers a means by which perceptions of rivers could be improved, the feel of an area changed, interest generated, and awareness of an issue raised. One element of the project was a youth art programme, which included the development of a short film to draw attention to the issue of plastic pollution (see Figure 20), the exhibition of work at a local art gallery, and murals that proved an effective way of making river corridor locations feel more welcoming (see Figure 21). Supported by acclaimed artists, young people were shown how to use their creativity to share powerful messages and make meaningful connections with rivers in the urban landscape. A student involved in the project described its impact:

"*Now I know more about the heritage of the River Don I feel I have a much deeper understanding and even a connection with the river*".

Participant surveys suggest that the Trust's wider community engagement activities are also having a positive effect on reconnecting people to the River Don:

"*Volunteering has made me feel connected to and proud of the place in which I live*"

"[*It motivates me to see*] *the value volunteering brings to the local community*"

"*Thanks for the incredible work you are doing to make Chesterfield a better place for the river life and for us humans*".

This has reinforced our belief that efforts to engage communities in our efforts to improve the ecology of the Don Catchment's urban rivers has worked synergistically, benefitting both local communities and the river ecosystem.

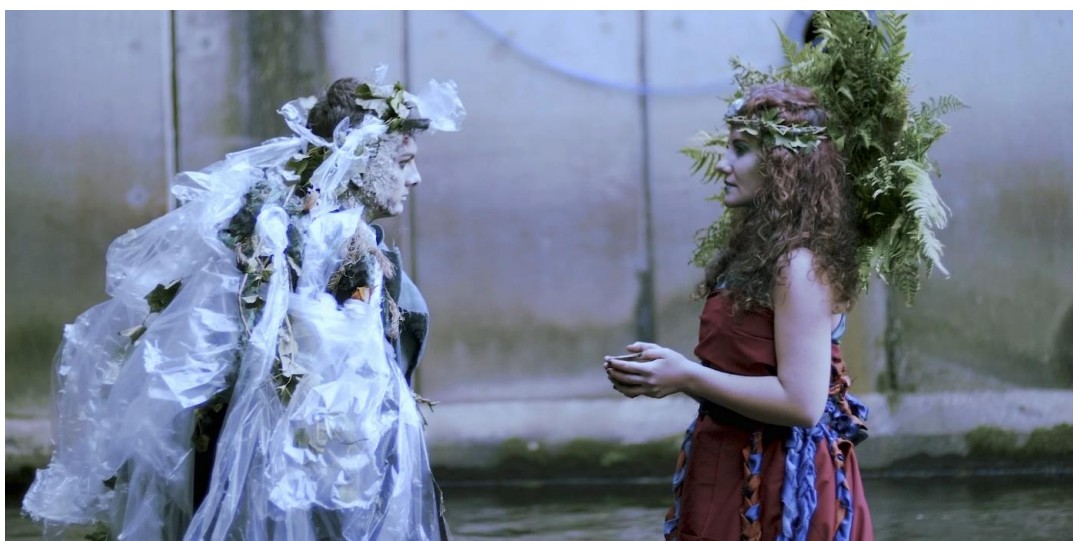

**Figure 20.** A still from the short film created by college students during 'The River Project: Categories of Life and Death', showing the personification of the River Don and exploring the themes of pollution and renewal. Credit: Eelyn Lee 2020.

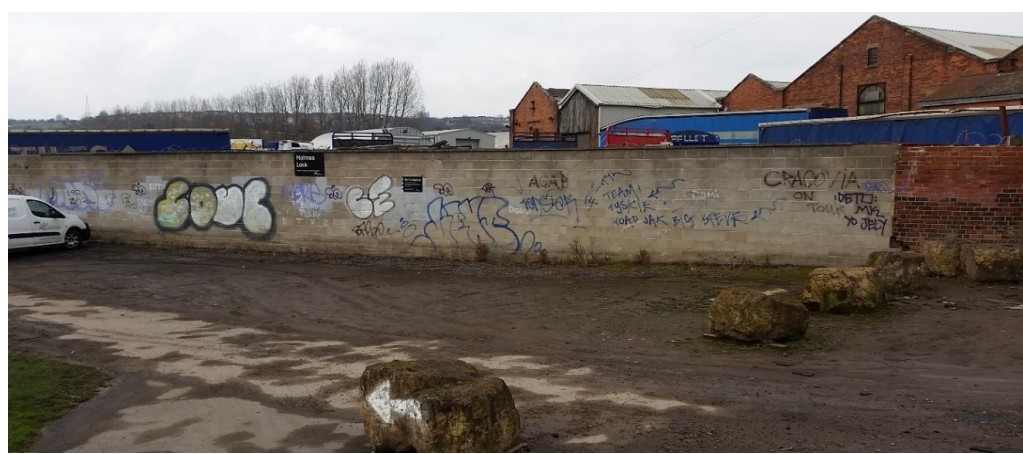

**(a)** a waterside wall with graffiti that creates an unwelcoming feel.

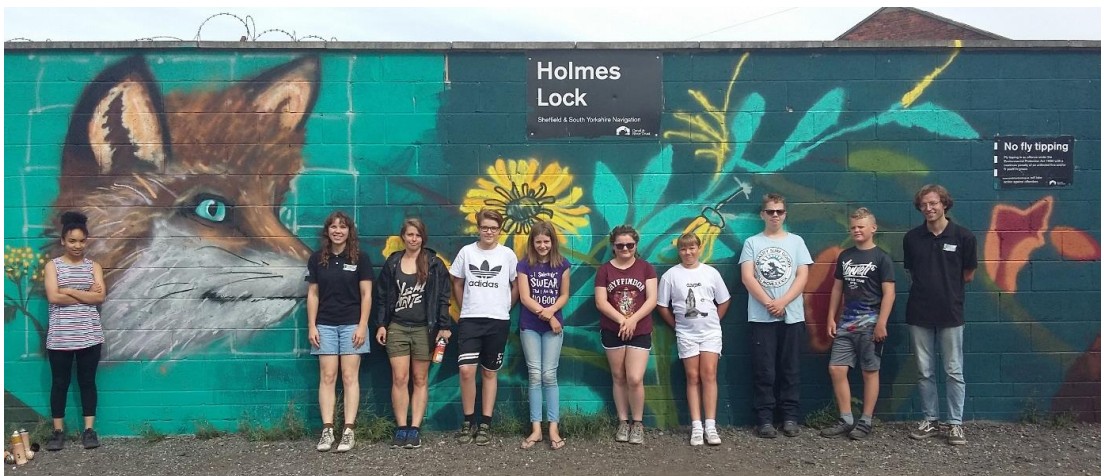

**(b)** the same wall after young people had created a mural under the guidance of local street artist Sarah Yates (also known as Faunagraphic).

**Figure 21.** (**a**) a waterside wall with graffiti that creates an unwelcoming feel. (**b**) the same wall after young people had created a mural under the guidance of local street artist Sarah Yates (also known as Faunagraphic). Photograph: DCRT 2018.

## 6. Discussion

In the scope of this special issue of Sustainability on urban rivers, Dempsey and Pattacini contend that 'urban design is highly dependent on contextual aspects', and call for 'contributions which explore cultural, social, and ecological dimensions of urban riverside landscapes'. Here we have supported their contention by describing how the unique history of urbanisation of the Don Catchment, and its industrial development in particular, has shaped the catchment's urban river corridors. This history has determined the expression of the 'urban river syndrome' in the catchment, which in turn has dictated the nature of the remedy, i.e., the type of projects that the Don Catchment Rivers Trust and other organisations are delivering to enhance the environmental, cultural and social value of the urban rivers.

The Don Catchment is set apart from many other catchments around the world by the early establishment of the area as a centre of industry, the extraordinary degree to which industry developed and expanded along the river valleys, and then the deindustrialisation of the 1970s and 1980s. The legacy of this history is manifested in numerous and sometimes distinctive ways. For example, the river network is highly fragmented by a high concentration of weirs, largely relicts from water-powered industry [19]. The rapid population growth, urbanisation and pollution issues in the nineteenth century led to the culverting of many streams and some rivers, something that would not normally be done in modern Britain. It also left much of Sheffield with an outdated combined sewer system [27].

The differing patterns and outcomes of urbanisation described in this paper demonstrate that the urban river corridors in the Don Catchment are heterogenous, and differ greatly not only between settlements, but also within them. This has had ramifications in terms of the opportunities for interventions in the urban river corridors, as exemplified by the case of green space. The dereliction of industry has facilitated the greening of the river corridors, something which has been embraced by the local authorities that recognise these continuous city spanning green elements as 'biohighways' that provide landscape-scale ecological connectivity both for freshwater and terrestrial species [12], as well as potential greenways for people. This vision has been realised to varying degrees in different places. In the Dearne Valley in Barnsley, industry retreated and was not replaced with new development, allowing the establishment of large areas of parkland. In the Don Valley downstream of Sheffield city centre, because formerindustrial sites were largely redeveloped with retail and light industry, there was less opportunity for the creation of public green space, though enough to enable the establishment of the riverside Five Weirs Walk and the inner-city Salmon Pastures Nature Reserve. In central Sheffield, there has been less demolition, and space is at more of a premium, so riverside walks and green spaces have been harder to establish, though this is gradually being achieved in an opportunistic piece-meal fashion, such as the creation of a pocket park on the Porter Brook in Sheffield in 2017 [34].

The interaction between cultural and social dimensions and the history of the urban rivers is an additional dynamic that distinguishes the catchment. The dreadful historic state of the rivers caused widespread negative attitudes to the rivers that still persist today, which is why we believe that perceptions of the rivers need to be improved if their potential as green-blue infrastructure is to be fully realised. Not all views are negative, however, and there are also plenty of positive narratives in the public discourse that celebrate the ecological recovery of the rivers in recent decades e.g., [33]. This positive celebratory narrative runs counter to a broader narrative in the UK, which focuses on concerns of lack of improvement or even declines in water quality [57].

There is also an aspect to British society that is notably pronounced compared to the global average, which is the strength of civil society. A wealth of civil society organisations have contributed and are continuing to contribute towards the improvement of the catchment's urban rivers. This has been successful due to the willingness of local government, public bodies and funders to engage with and support their activities. However, an important consideration is that civil society is strongest in areas with more social capital [58], which is one of the reasons why Sheffield, and more affluent communities in Sheffield, are

more advanced than other urban centres in terms of civil society led initiatives to improve river corridors.

While we have discussed aspects that make the Don Catchment's urban rivers distinctive within and outside of the UK, it should still be stressed that the pressures they face are not unique but are shared to some degree by many others around the world. Indeed, the response of river ecosystems to urbanisation can be fairly predictable. This is well demonstrated by considering the success that has been had in using indices based on aquatic macroinvertebrate communities as tools for assessing the anthropogenic pressures a river has been subjected to [59,60]. Rather, what creates the huge variation in the nature of urban rivers is the exact combination and relative magnitude of the anthropogenic pressures, the mechanisms that have caused them, the socio-cultural context, and other broader factors such as economics. We would argue that an appreciation of this is important to developing more sustainable urban river corridors.

Much has been achieved in the Don Catchment, but it is worth considering the scale of the challenge of attaining a more complete ecological recovery. If we look at the issue of habitat fragmentation caused by barriers, while 19 weirs on the Don have either had fish passes installed or have been removed, there are approximately 200 more weirs elsewhere in the catchment that still need to be addressed. Furthermore, little consideration has yet been given to the downstream migration of fish, which can also be inhibited by the presence of weirs. Weir notching is now recommended in some circumstances alongside fish pass construction as a solution to this problem. In addition, there are many other barriers in the catchment that fragment the river network, including culverts, pumping stations and outfalls.

Then there is the issue of habitat modification. Most of the rivers in the catchment are heavily modified in a variety of ways [12]. Lateral river connectivity (between the river and floodplain) has been almost completely severed, apart from during extreme flows, when flood defences are overwhelmed, or flood storage areas are allowed to fill. So far there has been some work to increase habitat quality in these rivers, but it represents <1% of the length of the river network.

Another big issue is that of water quality. Urban areas are an important source of diffuse pollution, from a wide range of contaminants including tyre wear and vehicle exhausts, chemicals that leach out of materials such as paint and plastics, any many other sources. As these pollutants are dispersed across urban surfaces and enter rivers via numerous flow pathways, then it will require much investment in interventions such as Sustainable Drainage Systems (SuDS) to intercept or treat urban runoff. Sheffield has made good progress in piloting the retro-fitting of SuDS, for example with the Grey to Green programme [61], but much more is needed. In addition, there are the combined-sewer overflows, which again require large amounts of investment to remedy. Altogether, these examples illustrate the point that the ecological restoration of the urban rivers in the Don Catchment will be a monumental task.

## 7. Conclusions

To maximise the effectiveness of efforts to improve urban river corridors, decision makers must be informed by a nuanced understanding of contextual aspects such as the wider catchment's history. In the Don Catchment, the fate of the urban rivers has been intertwined with the industrialisation of the region. While the broad patterns and effects of urbanisation are shared with catchments around the world, the specifics and extent to which problems, opportunities and solutions are manifested are distinctive to the Don Catchment. We have shown that the growth of industry resulted in a river network heavily fragmented by weirs, severe pollution that caused the ecological death of river ecosystems, and the disconnection of communities from stark urban river corridors. Widescale deindustrialisation in the 1970s and 1980s has resulted in a partial 'recombinant' ecological recovery of the rivers, and provided opportunity to develop urban river corridors into wildlife-rich blue-green spaces for local communities. This potential is dependent on

the location, as context matters between and within settlements, as the catchment's urban river corridors are very variable.

The history of the catchment has determined the issues we face today, and is the reason why the main activities of our organisation includes the addressing of weirs to restore ecological connectivity through the river network, Natural Flood Management, and encouraging communities to reengage with urban rivers so that their potential as green-blue infrastructure can be realised. Negative opinions towards rivers in the Don Catchment mean that river improvement is not just about dealing with the physical manifestation of urban river syndrome, it is also about winning hearts and minds and tackling deep-rooted perceptions of what a river should and can be. Despite the scale of the challenges, a strong and growing partnership of civil society, local government, public bodies, and the regional water company provides reason for optimism for further positive change in the catchment.

**Author Contributions:** Conceptualization, E.S., D.C., A.C., M.D., C.F., B.F., S.G., S.H., D.R., R.W. and P.W.; writing—original draft preparation, E.S., D.C., A.C., S.H., R.W. and P.W. All authors have read and agreed to the published version of the manuscript.

**Funding:** This research received no external funding.

**Institutional Review Board Statement:** Not applicable.

**Informed Consent Statement:** Not applicable.

**Data Availability Statement:** The datasets mapped in this study are openly available: South Yorkshire Historic Environment Characterisation data can be accesses at the UK Archaeology Data Service [https://doi.org/10.5284/1019858] (accessed on 26 May 2021). West Yorkshire Historic Landscape Characterisation data can be accesses at the UK Archaeology Data Service [https://doi.org/10.5284/1042125] (accessed on 26 May 2021). Derbyshire Historic Landscape Characterisation data can be accesses at the UK Archaeology Data Service [https://doi.org/10.5284/1039452] (accessed on 26 May 2021). The English Indices of Deprivation 2019 data is openly available [https://www.gov.uk/government/statistics/english-indices-of-deprivation-2019] (accessed on 26 May 2021).

**Conflicts of Interest:** The authors declare no conflict of interest.

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
