# Peer review of "Urban Rivers Corridors in the Don Catchment, UK: From Ignored, Ignoble and Industrial to Green, Seen and Celebrated"

_sustainability, doi:10.3390/su13147646_

Round 1

Reviewer 1 Report

Originality & Significance: This is a timely paper addressing a significant aspect of changing landscape management.  The significance of post-industrial landscapes and the urban planning options that are explored have assumed increasing concern as high-density over-development erases evidence of former history and localities. 

Quality of Presentation: This paper clearly explains the issues related to former industrial urban rivers and the specific catchment of the River Don, Sheffield. The catchment overview provides a clear, concise description of the basic facts as well as elucidating comparative changes using historical quotes.  This is well done given the constraints of word limits.

The structure of the paper enables the reader to recognise the landscape changes associated with the entire river dynamics and flows and the range of resulting industrial uses.   The environmental impact of these uses is also clearly explained.

This disciplined structure is maintained in second part of the paper, dealing with de-industrial land-uses and the continued negative community attitudes towards the river.   In terms of community attitudes, the paper would be improved by including the role of the artist in raising awareness about the emerging ecologies in wastelands. There are diverse examples of artist-led community engagement with post-industrial rivers, examples of which are described in chapter 6 https://books.apple.com/us/book/marginal-landscapes/id1146329496

https://www.academia.edu/32765308/MARGINAL_LANDSCAPES

The photographs communicate the issues and support the text; however, the date and photographer need to be acknowledged.

Interest & Overall Merit: This paper will be of interest to the readership of the Journal and hopefully will inspire further research in this area.

Author Response

Hello,

Thank you for your helpful comments.

We have actually used art in our projects, and so this aspect has made a nice addition to the paper (see lines 615 – 627).

Credits and years have been added to photo captions.

Best wishes

The authors

Reviewer 2 Report

This is an excellent piece of scholarship.  Indeed, I will employ a number of your examples in my courses.

I had few minor corrections/queries, that ought to be addressed.

Author Response

Hello,

Thank you for reading the manuscript and for your comments which we have addressed:

  1. We have added superscript formatting to units where this was missing.
  2. Given the reviewer’s question about the function of the waterpowered metal working sites we added the following sentence (Line 131):

Waterpower was harnessed for a variety of purposes including working bellows to blow air into forges, moving hammers, rollers, and cutters to shape metal, and turning grinding wheels to sharpen blades.

Best wishes

The authors

Reviewer 3 Report

 In the project, the environment around river side and water quality are successfully improved at the same time to keeping the historical heritages. The introduction of such fruitful project may be useful to the city project planner and river engineers. Several revises are necessary to make clear the meaning indicated in the text.

Revision Point;

  • Title : “Improvement of Urban Rivers Corridors …” is better? Please re-think about the title,
  • Figure 1 : How far is the boundary of right side of the figure from the sea(North Sea)?
  • Figure 4 : What do you mean by “River Sheaf.”?
  • Figure 15 : Please indicate the name of main city for the reference.

Author Response

Hello,

Thank you for reading the manuscript and for your comments. In response we have:

For figure 1, in the inset locator map we have labelled the North Sea and added a scale so that the reader can gain an impression of the distance between the sea and the Don Catchment.

In the caption for Figure 4 we explain that the River Sheaf is one of Sheffield’s main rivers

In Figure 15 we have increased the text size of Sheffield so that the reader can more easily recognise that it is the main city in the map

Regarding the title, we quite like it and would prefer to keep it if it is possible? We think that it encapsulates the paper in an lively and interesting way.

Best wishes

The authors

Reviewer 4 Report

Manuscript “Urban Rivers Corridors in the Don Catchment, UK: From Ignored, Ignoble and Industrial to Green, Seen and Celebrated” by Ed Shaw, Debbie Coldwell, Anthony Cox, Matt Duffy, Chris Firth, Beckie Fulton, Sue Goodship, Sally Hyslop, 4 David Rowley, Rachel Walker and Peter Worrall

The manuscript presents an interesting vision of a river ecosystem subject to the industrialization phase and the subsequent deindustrialization phase. I believe that this study on the quality and sustainability of the river system must be an essential part of the knowledge of all ecologists and urban planners. The problems, the issues, the urban river syndrome, are themes that constantly recur in western post-industrial societies and these aspects must be developed in an unified way among the various stakeholders in order to achieve a qualitative recovery objective in all the fields, from the environmental to the social ones.

With regard to the quality of the river ecosystem, I would also broaden the application of modern approach techniques such as bio-ecological traits or multimetric indices, in addition to the aspects on the fish component. To do this I would suggest inserting appropriate bibliographic references (at the end of sentence 686-689), see the following examples:

- Pallottini M., Cappelletti D., Fabrizi A., Gaino E., Goretti E., Selvaggi R., Cereghino R. (2017). Macroinvertebrate functional trait responses to chemical pollution in agricultural-industrial landscapes. River Research and Applications, 33: 505-513.

- Mondy C.P., Villeneuve B., Archaimbault V., Usseglio-Polatera P. (2012). A new macroinvertebrate-based multimetric index (I2M2) to evaluate ecological quality of French wadeable streams fulfilling the WFD demands: A taxonomical and trait approach. Ecological Indicators, 18: 452-467.

- lines 85-87: why take the Yangtze river (the longest river in Asia and the third longest in the world, after the Nile and the Amazon) for this comparison? Maybe it would have been more appropriate to make a comparison with the River Thames?

- why is the city of Rotherham not shown in figure 1, but is shown in figure 9?

Author Response

Hello,

Thank you for reading the manuscript and for your comments.

With regard to your suggestion on making reference to multimetric indices, I wasn’t entirely sure if I had understood correctly. In the paragraph 686-689 I added the text in italics, to make the point that such metrics demonstrate that urban river ecosystems do respond in predictable ways to anthropogenic pressures:

While we have discussed aspects that make the Don Catchment’s urban rivers distinctive within and outside of the UK, it should still be stressed that the pressures they face are not unique but are shared to some degree by many others around the world. Indeed, the response of the river ecosystems to urbanisation can be fairly predictable. This is well demonstrated by considering the success that has been had in using indices based on aquatic macroinvertebrate communities as tools for assessing the anthropogenic pressures a river has been subjected to [59,60]. Rather, what creates the huge variation in the nature of urban rivers is the exact combination and relative magnitude of the anthropogenic pressures, the mechanisms that have caused them, the socio-cultural context, and many other broader factors. We would argue that an appreciation of this is important to developing more sustainable urban river corridors. (lines 721-725)

I used the Yangtze as a comparator to the Don as it was at the other end of the spectrum in terms of width and flow. Its flow is greater than the Nile, and it was convenient as I knew offhand Nanjing was located on the river. However, on reflection I think it is of value to use a couple of other comparators of different sizes, so I have now added the Thames and the Rhine.  

I have now labelled Rotherham in Figure 1.

Best wishes

Ed